# The Pitfalls of Simplicity Bias in Neural Networks

**Harshay Shah**
Microsoft Research
harshay.rshah@gmail.com

**Kaustav Tamuly**
Microsoft Research
ktamuly2@gmail.com

**Aditi Raghunathan**
Stanford University
aditir@stanford.edu

**Prateek Jain**
Microsoft Research
prajain@microsoft.com

**Praneeth Netrapalli**
Microsoft Research
praneeth@microsoft.com

## Abstract

Several works have proposed Simplicity Bias (SB)—the tendency of standard training procedures such as Stochastic Gradient Descent (SGD) to find simple models—to justify why neural networks generalize well [1, 49, 74]. However, the precise notion of simplicity remains vague. Furthermore, previous settings [67, 24] that use SB to justify why neural networks generalize well do not simultaneously capture the non-robustness of neural networks—a widely observed phenomenon in practice [71, 36]. We attempt to reconcile SB and the superior standard generalization of neural networks with the non-robustness observed in practice by introducing piecewise-linear and image-based datasets, which (a) incorporate a precise notion of simplicity, (b) comprise multiple predictive features with varying levels of simplicity, and (c) capture the non-robustness of neural networks trained on real data. Through theoretical analysis and targeted experiments on these datasets, we make four observations:

(i) SB of SGD and variants can be extreme: neural networks can exclusively rely on the simplest feature and remain invariant to all predictive complex features.

(ii) The extreme aspect of SB *could* explain why seemingly benign distribution shifts and small adversarial perturbations significantly degrade model performance.

(iii) Contrary to conventional wisdom, SB can also hurt generalization on the same data distribution, as SB persists even when the simplest feature has less predictive power than the more complex features.

(iv) Common approaches to improve generalization and robustness—ensembles and adversarial training—can fail in mitigating SB and its pitfalls.

Given the role of SB in training neural networks, we hope that the proposed datasets and methods serve as an effective testbed to evaluate novel algorithmic approaches aimed at avoiding the pitfalls of SB.

## 1 Introduction

Understanding the superior generalization ability of neural networks, despite their high capacity to fit randomly labeled data [84], has been a subject of intense study. One line of recent work [67, 24] proves that linear neural networks trained with Stochastic Gradient Descent (SGD) on linearly separable data converge to the maximum-margin linear classifier, thereby explaining the superior generalization performance. However, maximum-margin classifiers are inherently robust to perturbations of data at prediction time, and this implication is at odds with concrete evidence that neural networks, in practice, are brittle to adversarial examples [71] and distribution shifts [52, 58, 44, 65]. Hence, the linear setting, while convenient to analyze, is insufficient to capture the non-robustness of neural networks trained on real datasets. Going beyond the linear setting, several works [1, 49, 74] argue that neural networks generalize well because standard training procedures have a bias towards learning

simple models. However, the exact notion of "simple" models remains vague and only intuitive. Moreover, the settings studied are insufficient to capture the brittleness of neural networks

Our goal is to formally understand and probe the *simplicity bias* (SB) of neural networks in a setting that is rich enough to capture known failure modes of neural networks and, at the same time, amenable to theoretical analysis and targeted experiments. Our starting point is the observation that on real-world datasets, there are several distinct ways to discriminate between labels (e.g., by inferring shape, color etc. in image classification) that are (a) predictive of the label to varying extents, and (b) define decision boundaries of varying complexity. For example, in the image classification task of white swans vs. bears, a linear-like "simple" classifier that only looks at color could predict correctly on most instances except white polar bears, while a nonlinear "complex" classifier that infers shape could have almost perfect predictive power. To systematically understand SB, we design modular synthetic and image-based datasets wherein different coordinates (or blocks) define decision boundaries of varying complexity. We refer to each coordinate / block as a *feature* and define a precise notion of feature *simplicity* based on the *simplicity of the corresponding decision boundary*.

**Proposed dataset.** Figure 1 illustrates a stylized version of the proposed synthetic dataset with two features, $\phi_1$ and $\phi_2$, that can perfectly predict the label with 100% accuracy, but differ in simplicity. The simplicity of a feature is precisely determined by the *minimum* number of linear pieces in the decision boundary that achieves optimal classification accuracy using that feature. For example, in Figure 1, the simple feature $\phi_1$ requires a linear decision boundary to perfectly predict the label, whereas complex feature $\phi_2$ requires four linear pieces. Along similar lines, we also introduce a collection of image-based datasets in which each image concatenates `MNIST` images (simple feature) and `CIFAR-10` images (complex feature). The proposed datasets, which incorporate features of varying predictive power and simplicity, allow us to systematically investigate and measure SB in SGD-trained neural networks.

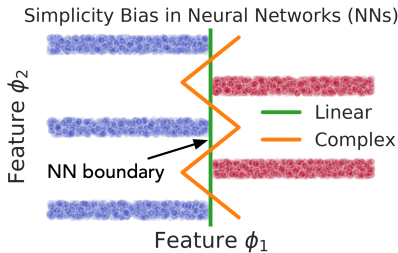

Simplicity Bias in Neural Networks (NNs)

Figure 1: Simple vs. complex features

**Observations from new dataset.** The ideal decision boundary that achieves high accuracy *and* robustness relies on all features to obtain a large margin (minimum distance from any point to decision boundary). For example, the orange decision boundary in Figure 1 that learns $\phi_1$ and $\phi_2$ attains 100% accuracy and exhibits more robustness than the linear boundary because of larger margin. Given the expressive power of large neural networks, one might expect that a network trained on the dataset in Figure 1 would result in the larger-margin orange piecewise linear boundary. However, in practice, we find quite the opposite—trained neural networks have a linear boundary. Surprisingly, neural networks exclusively use feature $\phi_1$ and remain *completely invariant* to $\phi_2$. More generally, we observe that SB is extreme: neural networks simply ignore several complex predictive features in the presence of few simple predictive features. We first theoretically show that one-hidden-layer neural networks trained on the piecewise linear dataset exhibit SB. Then, through controlled experiments, we validate the extreme nature of SB across model architectures and optimizers.

**Implications of extreme SB.** Theoretical analysis and controlled experiments reveal three major pitfalls of SB in the context of proposed synthetic and image-based datasets, which we *conjecture* to hold more widely across datasets and domains:

(i) Lack of robustness: Neural networks exclusively latch on to the simplest feature (e.g., background) at the expense of very small margin and completely ignore complex predictive features (e.g., semantics of the object), *even when all features have equal predictive power*. This results in susceptibility to small adversarial perturbations (due to small margin) and spurious correlations (with simple features). Furthermore, in Section 4, we provide a concrete connection between SB and data-agnostic and model-agnostic universal adversarial perturbations [47] observed in practice.

(ii) Lack of reliable confidence estimates: Ideally, a network should have high confidence only if all predictive features agree in their prediction. However due to extreme SB, the network has high confidence even if several complex predictive features contradict the simple feature, mirroring the widely reported inaccurate and substantially higher confidence estimates reported in practice [51, 25].

(iii) Suboptimal generalization: Surprisingly, neural networks exclusively rely on the simplest feature *even if it less predictive of the label than all complex features* in the synthetic datasets. Consequently, contrary to conventional wisdom, extreme SB can hurt robustness as well as generalization.

In contrast, prior works [8, 67, 24] only extol SB by considering settings where all predictive features are simple and hence do not reveal the pitfalls observed in real-world settings. While our results on the pitfalls of SB are established in the context of the proposed datasets, the two design principles underlying these datasets—combining multiple features of varying simplicity & predictive power and capturing multiple failure modes of neural networks in practice—suggest that our conclusions *could* be justifiable more broadly.

**Summary.** This work makes two key contributions. First, we design datasets that offer a precise stratification of features based on simplicity and predictive power. Second, using the proposed datasets, we provide theoretical and empirical evidence that neural networks exhibit extreme SB, which we postulate as a unifying contributing factor underlying key failure modes of deep learning: poor out-of-distribution performance, adversarial vulnerability and suboptimal generalization. To the best of our knowledge, prior works only focus on the positive aspect of SB: the lack of overfitting in practice. Additionally, we find that standard approaches to improve generalization and robustness—ensembles and adversarial training—do not mitigate simplicity bias and its shortcomings on the proposed datasets. Given the important implications of SB, we hope that the datasets we introduce serve (a) as a useful testbed for devising better training procedures and (b) as a starting point to design more realistic datasets that are amenable to theoretical analysis and controlled experiments.

**Organization.** We discuss related work in Section 2. Section 3 describes the proposed datasets and metrics. In Section 4, we concretely establish the extreme nature of Simplicity Bias (SB) and its shortcomings through theory and empirics. Section 5 shows that extreme SB can in fact hurt generalization as well. We conclude and discuss the way forward in Section 6.

## 2 Related Work

**Out-of-Distribution (OOD) performance**: Several works demonstrate that NNs tend to learn spurious features & low-level statistical patterns rather than semantic features & high-level abstractions, resulting in poor OOD performance [36, 21, 45, 52]. This phenomenon has been exploited to design backdoor attacks against NNs [6, 12] as well. Recent works [77, 76] that encourage models to learn higher-level features improve OOD performance, but require domain-specific knowledge to penalize reliance on spurious features such as image texture [21] and annotation artifacts [26] in vision & language tasks. Learning robust representations without domain knowledge, however, necessitates formalizing the notion of features and feature reliance; our work takes a step in this direction.

**Adversarial robustness**: Neural networks exhibit vulnerability to small adversarial perturbations [71]. Standard approaches to mitigate this issue—adversarial training [23, 42] and ensembles [69, 54, 37]—have had limited success on large-scale datasets. Consequently, several works have investigated reasons underlying the existence of adversarial examples: [23] suggests local linearity of trained NNs, [61] indicates insufficient data, [62] suggests inevitability in high dimensions, [9] suggests computational barriers, [16] proposes limitations of neural network architectures, and [33] proposes the presence of non-robust features. Additionally, Jacobsen et al. [34] show that NNs exhibit invariance to large label-relevant perturbations. Prior works have also demonstrated the existence of *universal adversarial perturbations* (UAPs) that are agnostic to model and data [55, 47, 73].

**Implicit bias of stochastic gradient descent** : Brutzkus et al. [8] show that neural networks trained with SGD provably generalize on linearly separable data. Recent works [67, 35] also analyze the limiting direction of gradient descent on logistic regression with linearly separable and non-separable data respectively. Empirical findings [49, 43] provide further evidence to suggest that SGD-trained NNs generalize well because SGD learns models of increasing complexity. Additional recent works investigate the implicit bias of SGD on non-linearly separable data for linear classifiers [35] and infinite-width two-layer NNs [13], showing convergence to maximum margin classifiers in appropriate spaces. In Section 4, we show that SGD's implicit bias towards simplicity can result in small-margin and feature-impoverished classifiers instead of large-margin and feature-dense classifiers.

**Feature reliance**: Two recent works study the relation between inductive biases of training procedures and the set of features that models learn. Hermann et al. [31] use color, shape and texture features in stylized settings to show that standard training procedures can (a) increase reliance on task-relevant features that are partially decodable using untrained networks and (b) suppress reliance on non-discriminative or correlated features. Ortiz et al. [53] show that neural networks learned using standard training (a) develop invariance to non-discriminative features and (b) adversarial training induces a sharp transition in the models' decision boundaries. In contrast, we develop a *precise* notion of feature simplicity and subsequently show that SGD-trained models can exhibit invariance to multiple *discriminative-but-complex* features. We also identify three pitfalls of this

|               | Accuracy          | AUC          | Logits               |
| ------------- | ----------------- | ------------ | -------------------- |
| S-Randomized  | 0.50              | 0.50         | randomly shuffled    |
| $S^c$-Randomized | standard accuracy | standard AUC | essentially identical |

Table 1: If the {S, $S^c$ }-randomized metrics of a model behave as above, then that model relies *exclusively* on S and is *invariant* to $S^c$.

phenomenon—poor OOD performance, adversarial vulnerability, suboptimal generalization—and show that adversarial training and standard ensembles do not mitigate the pitfalls of simplicity bias.

Multiple works mentioned above (a) differentially characterize *learned* features and *desired* features—statistical regularities vs. high-level concepts [36], syntactic cues vs. semantic meaning [45], robust vs. non-robust features [33]—and (b) posit that the mismatch between these features results in non-robustness. In contrast, our work probes *why* neural networks prefer one set of features over another and unifies the aforementioned feature characterizations through the lens of feature *simplicity*.

## 3 Preliminaries: Setup and Metrics

**Setting and metrics**: We focus on binary classification. Given samples $\widehat{\mathcal{D}} = \{(x_i, y_i)\}_{i=1}^n$ from distribution $\mathcal{D}$ over $\mathbb{R}^d \times \{-1, 1\}$, the goal is to learn a scoring function $s(x) : \mathbb{R}^d \to \mathbb{R}$ (such as logits), and an associated classifier $f : \mathbb{R}^d \to \{-1, 1\}$ defined as $f(x) = 2h(x)-1$ where $h(x) = \mathbb{1}\{\text{softmax}(s(x)) < 0.5\}$. We use two well-studied metrics for generalization and robustness:

**Standard accuracy.** The standard accuracy of a classifier $f$ is: $\mathbb{E}_{\mathcal{D}}\left[\mathbb{1}\{f(x) = y\}\right]$.

**$\delta$-Robust accuracy.** Given norm $\|\cdot\|$ and perturbation budget $\delta$, the $\delta$-robust accuracy of a classifier $f$ is: $\mathbb{E}_{\mathcal{D}}\left[\min_{\|\hat{x}-x\| \leq \delta} \mathbb{1}\{f(\hat{x}) = y\}\right]$.

Next, we introduce two metrics that quantitatively capture the extent to which a model relies on different input coordinates (or features). Let $S$ denote some subset of coordinates $[d]$ and $\overline{\mathcal{D}}^S$ denote the $S$-randomized distribution, which is obtained as follows: given $\mathcal{D}^S$, the marginal distribution of $S$, $\overline{\mathcal{D}}^S$ independently samples $((x^S, x^{S^c}), y) \sim \mathcal{D}$ and $\overline{x}^S \sim \mathcal{D}^S$ and then outputs $((\overline{x}^S, x^{S^c}), y)$. In $\overline{\mathcal{D}}^S$, the coordinates in $S$ are rendered independent of the label $y$. The two metrics are as follows.

**Definition 1** (Randomized accuracy)**.** Given data distribution $\mathcal{D}$, and subset of coordinates $S \subseteq [d]$, the $S$-randomized accuracy of a classifier $f$ is given by: $\mathbb{E}_{\overline{\mathcal{D}}^S}\left[\mathbb{1}\{f(x) = y\}\right]$.

**Definition 2** (Randomized AUC)**.** Given data distribution $\mathcal{D}$ and subset of coordinates $S \subseteq [d]$, the $S$-randomized AUC of classifier $f$ equals the area under the precision-recall curve of distribution $\overline{\mathcal{D}}^S$.

Our experiments use {S, $S^c$ }-randomized metrics—accuracy, AUC, logits—to establish that $f$ depends *exclusively on some features* S and *remains invariant to the rest* $S^c$. First, if (a) S-randomized accuracy and AUC equal 0.5 and (b) S-randomized logit distribution is a random shuffling of the original distribution (i.e., logits in the original distribution are randomly shuffled across true positives and true negatives), then $f$ depends *exclusively* on S. Conversely, if (a) $S^c$-randomized accuracy and AUC are equal to standard accuracy and AUC and (b) $S^c$-randomized logit distribution is essentially identical to the original distribution, then $f$ is *invariant* to $S^c$; Table 1 summarizes these observations.

### 3.1 Datasets

**One-dimensional Building Blocks**: Our synthetic datasets use three one-dimensional data blocks—linear, noisy linear and $k$-slabs—shown in top row of Figure 2. In the **linear** block, positive and negative examples are uniformly distributed in $[0.1, 1]$ and $[-1, -0.1]$ respectively. In the **noisy linear** block, given a noise parameter $p \in [0, 1]$, $1 - p$ fraction of points are distributed like the linear block described above and $p$ fraction of the examples are uniformly distributed in $[-0.1, 0.1]$. In $k$-**slab** blocks, positive and negative examples are distributed in $k$ well-separated, alternating regions.

**Simplicity of Building Blocks**: Linear classifiers can attain the optimal (Bayes) accuracy of 1 and $1 - {}^p/_2$ on the linear and $p$-noisy linear blocks respectively. For $k$-slabs, however, $(k-1)$-piecewise linear classifiers are required to obtain the optimal accuracy of 1. Consequently, the building blocks have a natural notion of simplicity: *minimum number of pieces required by a piecewise linear classifier to attain optimal accuracy*. With this notion, the linear and noisy linear blocks are simpler than $k$-slab blocks when $k > 2$, and $k$-slab blocks are simpler than $\ell$-slab blocks when $k < \ell$.

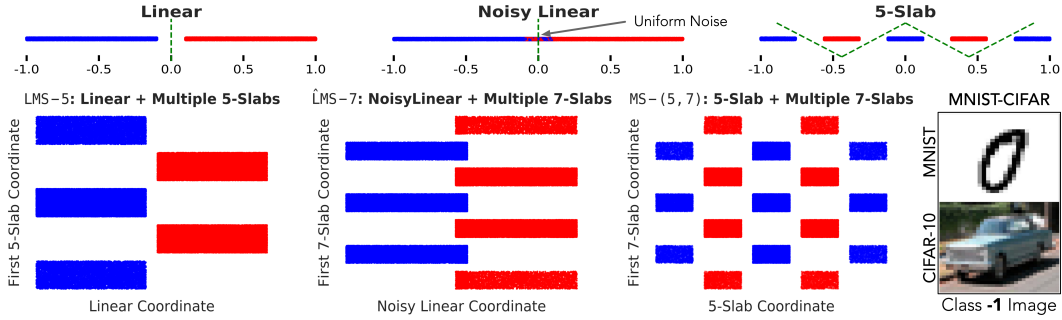

Figure 2: (Synthetic & Image-based Datasets) One-dimensional building blocks (top row)—linear, noisy linear, $k$-slab—are used to construct multi-dimensional datasets (bottom row): `LMS-5` (linear & multiple 5-slabs), $\hat{\texttt{LMS}}\texttt{-5}$ (noisy linear & multiple 5-slabs) and `MS-(5,7)` (5-slab & multiple 7-slabs). `MNIST-CIFAR` data vertically concatenates `MNIST` and `CIFAR` images (see Section 3.1).

**Multi-dimensional Synthetic Datasets**: We now outline four $d$-dimensional datasets wherein each coordinate corresponds to one of three building blocks described above. See Figure 2 for illustration.

- `LMS-k`: Linear and multiple $k$-slabs; the first coordinate is a linear block and the remaining $d-1$ coordinates are independent $k$-slab blocks; we use `LMS-5` & `LMS-7` datasets in our analysis.
- $\hat{\texttt{LMS}}\texttt{-k}$: Noisy linear and multiple $k$-slab blocks; the first coordinate is a *noisy* linear block and the remaining $d-1$ coordinates are independent $k$-slab blocks. The noise parameter $p$ is $0.1$ by default.
- `MS-(5,7)`: 5-slab and multiple 7-slab blocks; the first coordinate is a 5-slab block and the remaining $d-1$ coordinates are independent 7-slab blocks, as shown in Figure 2.
- `MS-5`: Multiple 5-slab blocks; all coordinates are independent 5-slab blocks.

We now describe the `LSN` (linear, 3-slab & noise) dataset, a stylized version of `LMS-k` that is amenable to theoretical analysis. We note that recent works [53, 20] empirically analyze variants of the `LSN` dataset. In `LSN`, conditioned on the label $y$, the first and second coordinates of $x$ are *singleton* linear and 3-slab blocks: linear and 3-slab blocks have support on $\{-1, 1\}$ and $\{-1, 0, 1\}$ respectively. The remaining coordinates are standard gaussians and not predictive of the label.

The synthetic datasets comprise features of varying simplicity; in `LMS-k`, $\hat{\texttt{LMS}}\texttt{-k}$, and `MS-(5,7)`, the first coordinate is the simplest feature and in `MS-5`, all features are equally simple. All datasets, even $\hat{\texttt{LMS}}\texttt{-k}$, can be *perfectly* classified via piecewise linear classifiers. Though the $k$-slab features are special cases of linear periodic functions on which gradient-based methods have been shown to fail for large $k$ [64], we note that we use small values of $k \in \{5, 7\}$ which are quickly learned by SGD in practice. Note that we (a) apply a random rotation matrix to the data and (b) use 50-dimensional synthetic data (i.e., $d = 50$) by default. Note that all code and datasets are available at the following repository: https://github.com/harshays/simplicitybiaspitfalls.

**MNIST-CIFAR Data**: The `MNIST-CIFAR` dataset consists of two classes: images in class $-1$ and class $1$ are vertical concatenations of `MNIST` digit zero & `CIFAR-10` automobile and `MNIST` digit one & `CIFAR-10` truck images respectively, as shown in Figure 2. The training and test datasets comprise 50,000 and 10,000 images of size $3 \times 64 \times 32$. The `MNIST-CIFAR` dataset mirrors the structure in the synthetic `LMS-k` dataset—both incorporate simple and complex features. The `MNIST` and `CIFAR` blocks correspond to the linear and $k$-slab blocks in `LMS-k` respectively. Also note that `MNIST` images are zero-padded & replicated across three channels to match `CIFAR` dimensions before concatenation.

Appendix B provides details about the datasets, models, and optimizers used in our experiments. In Appendix C, we show that our results are robust to the exact choice of `MNIST-CIFAR` class pairs.

# 4 Simplicity Bias (SB) is Extreme and Leads to Non-Robustness

We first establish the *extreme* nature of SB in neural networks (NNs) on the proposed synthetic datasets using SGD and variants. In particular, we show that for the datasets considered, *if all features have full predictive power, NNs rely exclusively on the simplest feature* $\mathsf{S}$ *and remain invariant to all complex features* $\mathsf{S}^c$. Then, we explain why extreme SB on these datasets results in neural networks that are vulnerable to distribution shifts and data-agnostic & transferable adversarial perturbations.

## 4.1 Neural networks provably exhibit Simplicity Bias (SB)

We consider the `LSN` dataset (described in Section 3.1) that has one *linear* coordinate and one *3-slab* coordinate, both fully predictive of the label on their own; the remaining $d{-}2$ noise coordinatesI do not have any predictive power. Now, a "large-margin" one-hidden-layer NN with ReLU activation should give equal weight to the linear and 3-slab coordinates. However, we prove that NNs trained with standard mini-batch gradient descent (GD) on the `LSN` dataset (described in Section 3.1) provably learns a classifier that *exclusively* relies on the "simple" linear coordinate, thus exhibiting simplicity bias at the cost of margin. Further, our claim holds even when the margin in the linear coordinate (minimum distance between linear coordinate of positives and negatives) is significantly smaller than the margin in the slab coordinate. The proof of the following theorem is presented in Appendix F.

**Theorem 1.** *Let $f(x) = \sum_{j=1}^{k} v_j \cdot ReLU(\sum_{i=1}^{d} w_{i,j} x_i)$ denote a one-hidden-layer neural network with $k$ hidden units and ReLU activations. Set $v_j = \pm 1/\sqrt{k}$ w.p. $1/2 \ \forall j \in [k]$. Let $\{(x^i, y^i)\}_{i=1}^{m}$ denote i.i.d. samples from `LSN` where $m \in [cd^2, d^\alpha/c]$ for some $\alpha > 2$. Then, given $d > \Omega(\sqrt{k}\log k)$ and initial $w_{ij} \sim \mathcal{N}(0, \frac{1}{dk \log^4 d})$, after $O(1)$ iterations, mini-batch gradient descent (over $w$) with hinge loss, step size $\eta = \Omega(\log d)^{-1/2}$, mini-batch size $\Theta(m)$, satisfies:*
- *Test error is at most $1/poly(d)$*
- *The learned weights of hidden units $w_{ij}$ satisfy:*

$$\underbrace{|w_{1j}| = \frac{2}{\sqrt{k}} \left(1 - \frac{c}{\sqrt{\log d}}\right) + O\left(\frac{1}{\sqrt{dk}\log d}\right)}_{\textbf{\textit{Linear Coordinate}}}, \ \underbrace{|w_{2,j}| = O\left(\frac{1}{\sqrt{dk}\log d}\right)}_{\textbf{\textit{3-Slab Coordinate}}}, \ \underbrace{\|w_{3:d,j}\| = O\left(\frac{1}{\sqrt{k}\log d}\right)}_{d-2 \ \textbf{\textit{Noise Coordinates}}}$$

*with probability greater than $1 - \frac{1}{poly(d)}$. Note that $c$ is a universal constant.*

**Remarks**: First, we see that the trained model essentially relies only on the linear coordinate $w_{1j}$—SGD sets the value of $w_{1j}$ roughly $\tilde{\Omega}(\sqrt{d})$ larger than the slab coordinates $w_{2j}$ that do not change much from their initial value. Second, the initialization we use is widely studied in the deep learning theory [46, 79] as it better reflects the practical performance of neural networks [14]. Third, given that LSN is linearly separable, Brutzkus et al. [8] also guarantee convergence of test error. However, our result additionally gives a precise description of the *final* classifier. Finally, we note that our result shows that extreme SB bias holds even for overparameterized networks ($k = O(d^2/\log d)$).

## 4.2 Simplicity Bias (SB) is Extreme in Practice

We now establish the extreme nature of SB on datasets with features of varying simplicity—`LMS-5`, `MS-(5,7)`, `MNIST-CIFAR` (described in Section 3)—across *multiple model architectures and optimizers*. Recall that (a) the simplicity of one-dimensional building blocks is defined as the number of pieces required by a piecewise linear classifier acting *only* on that block to get optimal accuracy and (b) `LMS-5` has one linear block & multiple 5-slabs, `MS-(5,7)` has one 5-slab and multiple 7-slabs and `MNIST-CIFAR` concatenates `MNIST` and `CIFAR10` images. We now use `S` to denote the simplest feature in each dataset: linear in `LMS-5`, 5-slab in `MS-(5,7)`, and `MNIST` in `MNIST-CIFAR`.

We first consistently observe that SGD-trained models trained on `LMS-5` and `MS-(5,7)` datasets exhibit extreme SB: they *exclusively* rely on the simplest feature `S` and remain invariant to all complex features $S^c$. Using `S`-randomized & $S^c$-randomized metrics summarized in Table 1, we first establish extreme SB on fully-connected (FCN), convolutional (CNN) & sequential (GRU [15]) models. We observe that the `S`-randomized AUC is 0.5 across models. That is, unsurprisingly, all models are critically dependent on `S`. Surprisingly, however, $S^c$-randomized AUC of all models on both datasets equals 1.0. That is, arbitrarily perturbing $S^c$ coordinates has *no impact* on the class predictions or the ranking of true positives' logits against true negatives' logits. One might expect that perturbing $S^c$ would at least bring the *logits* of positives and negatives closer to each other. Figure 3(a) answers this in negative—the logit distributions over true positives of `(100,1)-FCNs` (i.e., with width 100 & depth 1) remain unchanged even after randomizing *all* complex features $S^c$. Conversely, randomizing the simplest feature `S` randomly shuffles the original logits across true positives as well as true negatives. The two-dimensional projections of `FCN` decision boundaries in Figure 3(c) visually confirm that `FCN`s exclusively depend on the simpler coordinate `S` and are invariant to all complex features $S^c$.

Note that sample size and model architecture do not present any obstacles in learning complex features $S^c$ to achieve 100% accuracy. In fact, if `S` is *removed* from the dataset, SGD-trained models

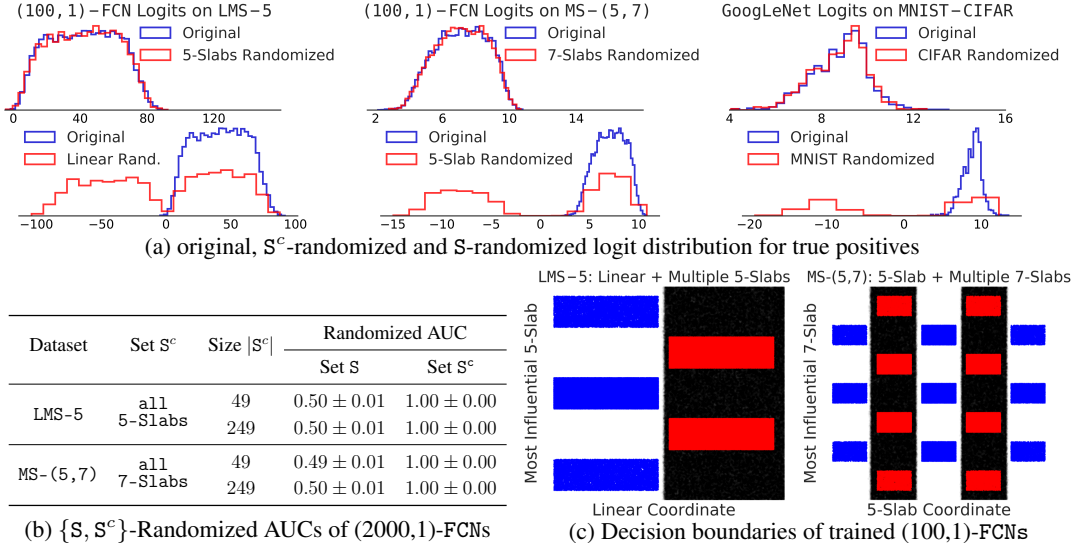

(a) original, $S^c$-randomized and $S$-randomized logit distribution for true positives

| Dataset | Set $S^c$ | Size $\|S^c\|$ | Randomized AUC | |
|---|---|---|---|---|
| | | | Set $S$ | Set $S^c$ |
| LMS-5 | all | 49 | $0.50 \pm 0.01$ | $1.00 \pm 0.00$ |
| | 5-Slabs | 249 | $0.50 \pm 0.01$ | $1.00 \pm 0.00$ |
| MS-(5,7) | all | 49 | $0.49 \pm 0.01$ | $1.00 \pm 0.00$ |
| | 7-Slabs | 249 | $0.50 \pm 0.01$ | $1.00 \pm 0.00$ |

(b) $\{S, S^c\}$-Randomized AUCs of (2000,1)-FCNs

(c) Decision boundaries of trained (100,1)-FCNs

Figure 3: Extreme SB on LMS-5, MS-(5,7) and MNIST-CIFAR datasets (a) $S$-randomized logit distribution of true positives is essentially identical to the original logit distribution of true positives (before randomization). However, $S^c$-randomized logit distribution of true positives is a randomly shuffled version of original logit distribution; $S^c$-randomized logits are *shuffled across true positives and negatives*. (b) $\{S, S^c\}$-randomized AUCs (summarized in Table 1) are $0.5$ and $1.0$ respectively for varying number of complex features $\|S^c\|$. (c) FCN decision boundaries projected onto $S$ & the most influential coordinate in $S^c$ shows that the boundary depends only on $S$ and is invariant to $S^c$.

with the same sample size indeed rely on $S^c$ to attain 100% accuracy. Increasing the number of complex features does not mitigate extreme SB either. Figure 3(b) shows that even when there are 249 complex features and only one simple feature, (2000,1)-FCNs exclusively rely on the simplest feature $S$; randomizing $S^c$ keeps AUC score of $1.0$ intact but simply randomizing $S$ drops the AUC score to $0.5$. (2000,1)-FCNs exhibit extreme SB despite their expressive power to learn large-margin classifiers that rely on all simple *and* complex features.

Similarly, on the MNIST-CIFAR dataset, MobileNetV2 [60], GoogLeNet [70], ResNet50 [27] and DenseNet121 [32] exhibit extreme SB. All models exclusively latch on to the simpler MNIST block to acheive 100% accuracy and remain invariant to the CIFAR block, even though the CIFAR block alone is almost fully predictive of its label—GoogLeNet attains 95.4% accuracy on the corresponding CIFAR binary classification task. Figure 3(a) shows that randomizing the simpler MNIST block randomly shuffles the logit distribution of true positives whereas randomizing the CIFAR block has no effect—the CIFAR-randomized and original logit distribution over true positives essentially overlap.

To summarize, we use $S$-randomized and $S^c$-randomized metrics to establish that models trained on synthetic and image-based datasets exhibit extreme SB: *If all features have full predictive power, NNs rely exclusively on the simplest feature $S$ and remain invariant to all complex features $S^c$.* We further validate our results on extreme SB across model architectures, activation functions, optimizers and regularization methods such as $\ell_2$ regularization and dropout in Appendix C.

### 4.3 Extreme Simplicity Bias (SB) leads to Non-Robustness

Now, we discuss how our findings about extreme SB in Section 4.2 can help reconcile poor OOD performance and adversarial vulnerability with superior generalization on the same data distribution.

**Poor OOD performance**: Given that neural networks tend to heavily rely on spurious features [45, 52], state-of-the-art accuracies on large and diverse validation sets provide a false sense of security; even benign distributional changes to the data (e.g., domain shifts) during prediction time can drastically degrade or even nullify model performance. This phenomenon, though counter-intuitive, can be easily explained through the lens of extreme SB. Specifically, we hypothesize that spurious features are *simple*. This hypothesis, when combined with extreme SB, explains the outsized impact of spurious features. For example, Figure 3(b) shows that simply perturbing the simplest (and

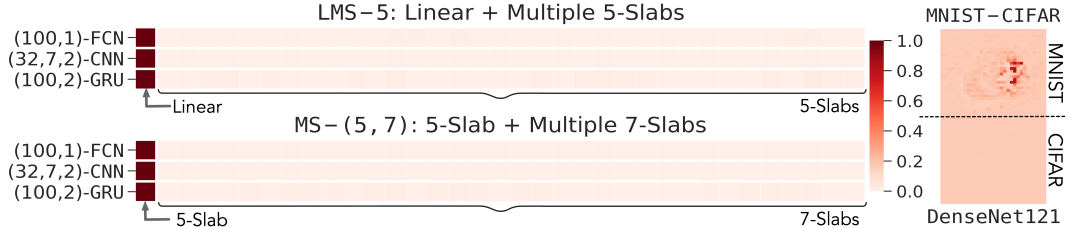

Figure 4: Extreme SB results in vulnerability to small-magnitude model-agnostic and data-agnostic Universal Adversarial Perturbations (UAPs) that nullify model performance by only perturbing the simplest feature S. $\ell_2$ UAPs utilize most of the perturbation budget to attack S alone: 99.6% for linear in LMS-5, 99.9% for 5-slab in MS-(5,7) and 99.3% for MNIST pixels in MNIST-CIFAR. The UAPs in this figure are rescaled for visualization purposes.

potentially spurious in practice) feature S drops the AUC of trained neural networks to 0.5, thereby nullifying model performance. Randomizing *all* complex features $S^c$—5-slabs in LMS-5, 7-slabs in MS-(5,7), CIFAR block in MNIST-CIFAR—has negligible effect on the trained neural networks— $S^c$-randomized and original logits essentially overlap—even though $S^c$ and S have equal predictive power This further implies that approaches [29, 40] that aim to detect distribution shifts based on model outputs such as logits or softmax probabilities may themselves fail due to extreme SB.

**Adversarial Vulnerability**: Consider a classifier $f^*$ that attains 100% accuracy on the LMS-5 dataset by taking an average of the linear classifier on the linear coordinate and $d-1$ piecewise linear classifiers, one for every 5-slab coordinate. By relying on all $d$ features, $f^*$ has $\mathcal{O}(\sqrt{d})$ margin. Now, given the large margin, $f^*$ also attains high robust accuracy to $\ell_2$ adversarial perturbations that have norm $\mathcal{O}(\sqrt{d})$—the perturbations need to attack at least $\Omega(d)$ coordinates to flip model predictions. However, despite high model capacity, SGD-trained NNs do not learn robust and large-margin classifiers such as $f^*$. Instead, due to extreme SB, SGD-trained NNs exclusively rely on the simplest feature S. Consequently, $\ell_2$ perturbations with norm $\mathcal{O}(1)$ are enough to flip predictions and degrade model performance. We validate this hypothesis in Figure 4, where FCNs, CNNs and GRUs trained on LMS-5 and MS-(5,7) as well as DenseNet121 trained on MNIST-CIFAR are vulnerable to small universal (i.e., data-agnostic) adversarial perturbations (UAPs) of the simplest feature S. For example, Figure 4 shows that the $\ell_2$ UAP of DenseNet121 on MNIST-CIFAR only attacks a few pixels in the simpler MNIST block and does not perturb the CIFAR block. Extreme SB also explains why data-agnostic UAPs of one model transfer well to another: the notion of simplicity is consistent across models; Figure 4 shows that FCNs, CNNs and GRUs trained on LMS-5 and MS-(5,7) essentially learn the same UAP. Furthermore, invariance to complex features $S^c$ (e.g., CIFAR block in MNIST-CIFAR) due to extreme SB explains why "natural" [30] and semantic label-relevant perturbations [5] that modify the true image class do not alter model predictions.

To summarize, through theoretical analysis and extensive experiments on synthetic and image-based datasets, we (a) establish that SB is extreme in nature across model architectures and datasets and (b) show that extreme SB can result in poor OOD performance and adversarial vulnerability, *even when all simple and complex features have equal predictive power*.

## 5 Extreme Simplicity Bias (SB) can hurt Generalization

In this section, we show that, contrary to conventional wisdom, extreme SB can potentially result in suboptimal generalization of SGD-trained models on the same data distribution as well. This is

| Accuracy | (100,1)-FCN | (200,1)-FCN | (300,1)-FCN | (100,2)-FCN | (200,2)-FCN | (300,2)-FCN |
|---|---|---|---|---|---|---|
| Training Data | $0.984 \pm 0.003$ | $0.998 \pm 0.000$ | $0.995 \pm 0.000$ | $0.999 \pm 0.000$ | $0.997 \pm 0.002$ | $0.998 \pm 0.002$ |
| Test Data | $0.940 \pm 0.002$ | $0.949 \pm 0.003$ | $0.948 \pm 0.002$ | $0.945 \pm 0.003$ | $0.946 \pm 0.003$ | $0.947 \pm 0.002$ |
| $S^c$-Randomized | $0.941 \pm 0.001$ | $0.946 \pm 0.001$ | $0.946 \pm 0.001$ | $0.944 \pm 0.001$ | $0.945 \pm 0.001$ | $0.946 \pm 0.001$ |
| S-Randomized | $0.498 \pm 0.001$ | $0.498 \pm 0.000$ | $0.497 \pm 0.001$ | $0.498 \pm 0.001$ | $0.497 \pm 0.000$ | $0.498 \pm 0.001$ |

Table 2: Extreme SB can hurt generalization: FCNs of depth $\{1, 2\}$ and width $\{100, 200, 300\}$ trained on L̂MS-7 data with SGD attain 95% test accuracy. The randomized accuracies show that FCNs exclusively rely on the simpler noisy linear feature S and remain invariant to all 7-slab features that have 100% predictive power.

because exclusive reliance on the simplest feature S can persist even when *every complex feature in* $S^c$ *has significantly greater predictive power than* S.

We verify this phenomenon on L̂MS-7 data defined in Section 3. Recall that L̂MS-7 has one noisy linear coordinate S with 95% predictive power (i.e., 10% noise in linear coordinate) and multiple 7-slab coordinates $S^c$, each with 100% predictive power. Note that our training sample size is large enough for FCNs of depth $\{1, 2\}$ and width $\{100, 200, 300\}$ trained on $S^c$ *only* (i.e., after removing S from data) to attain 100% test accuracy. However, when trained on L̂MS-7 (i.e., including S), SGD-trained FCNs exhibit extreme SB and *only* rely on S, the noisy linear coordinate. In Table 2, we report accuracies of SGD-trained FCNs that are selected based on validation accuracy after performing a grid search over four SGD hyperparameters: learning rate, batch size, momentum, and weight decay. The train, test and randomized accuracies in Table 2 collectively show that FCNs exclusively rely on the noisy linear feature and consequently attain 5% generalization error.

To summarize, the mere presence of a simple-but-noisy feature in L̂MS-7 data can significantly degrade the performance of SGD-trained FCNs due to extreme SB. Note that our results show that even an extensive grid search over SGD hyperparameters does not improve the performance of SGD-trained FCNs on L̂MS-7 data but does not necessarily imply that mitigating SB via SGD and its variants is impossible. We provide additional information about the experiment setup in Appendix D.

## 6    Conclusion and Discussion

We investigated Simplicity Bias (SB) in SGD-trained neural networks (NNs) using synthetic and image-based datasets that (a) incorporate a precise notion of feature simplicity, (b) are amenable to theoretical analysis and (c) capture the non-robustness of NNs observed in practice. We first showed that one-hidden-layer ReLU NNs provably exhibit SB on the LSN dataset. Then, we analyzed the proposed datasets to empirically demonstrate that SB can be extreme, and can help explain poor OOD performance and adversarial vulnerability of NNs. We also showed that, contrary to conventional wisdom, extreme SB can potentially hurt generalization.

**Can we mitigate SB?** It is natural to wonder if any modifications to the standard training procedure can help in mitigating extreme SB and its adverse consequences. In Appendix E, we show that well-studied approaches for improving generalization and adversarial robustness—ensemble methods and adversarial training—do not mitigate SB, at least on the proposed datasets. Specifically, "vanilla" ensembles of independent models trained on the proposed datasets mitigate SB to some extent by aggregating predictions, but continue to exclusively rely on the simplest feature. That is, the resulting ensemble remains *invariant* to *all* complex features (e.g., 5-Slabs in LMS-5). Our results suggest that in practice, the improvement in generalization due to vanilla ensembles stem from combining multiple simple-but-noisy features (such as color, texture) and not by learning diverse and complex features (such as shape). Similarly, adversarially training FCNs on the proposed datasets increases margin (and hence adversarial robustness) to some extent by combining multiple simple features but does not achieve the maximum possible adversarial robustness; the resulting adversarially trained models remain invariant to *all* complex features.

Our results collectively motivate the need for novel algorithmic approaches that avoid the pitfalls of extreme SB. Furthermore, the proposed datasets capture the key aspects of training neural networks on real world data, while being amenable to theoretical analysis and controlled experiments, and can serve as an effective testbed to understand deep learning phenomena. evaluating new algorithmic approaches aimed at avoiding the pitfalls of SB.

## Broader Impact

Our work is foundational in nature and seeks to improve our understanding of neural networks. We do not foresee any significant societal consequences in the short term. However, in the long term, we believe that a concrete understanding of deep learning phenomena is essential to develop reliable deep learning systems for practical applications that have societal impact.

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
