[Supplementary Material]

# Appendices

The supplementary material is organized as follows. We first discuss additional related work and provide experiment details in Section 2 and Appendix B respectively. In Appendix C, we provide additional experiments to further validate the extreme nature of Simplicity Bias (SB). Then, in Appendix D, we provide additional information about the experiment setup used to to show that extreme SB can hurt generalization. We evaluate the extent to which ensemble methods and adversarial training mitigate Simplicity Bias (SB) in Appendix E. Finally, we provide the proof of Theorem 1 in Appendix F.

## A   Additional Related Work

In this section, we provide a more thorough discussion of relevant work related to margin-based generalization bounds, adversarial attacks and robustness, and out-of-distribution (OOD) examples.

**Margin-based generalization bounds**: Building up on the classical work of [3], recent works try to obtain tighter generalization bounds for neural networks in terms of *normalized* margin [4, 50, 18, 22]. Here, margin is defined as the difference in the probability of the true label and the largest probability of the incorrect labels. While these bounds seem to capture generalization of neural networks at a coarse level, it has been argued [48] that these approaches may be incapable of fully explaining the generalization ability of neural networks. Furthermore, it is unclear if the notion of model complexity used in these works, based on Lipschitz constant, captures generalization ability accurately. In any case, our results suggest that due to extreme simplicity bias (SB), even if a formulation captures both margin and model complexity accurately, current optimization techniques may not be able to find the optimal solution in terms of generalization *and* robustness-, as they are strongly biased towards small-margin classifiers that exclusively rely on the simplest features.

**Adversarial Defenses**: Neural networks trained using standard procedures such as SGD are extremely vulnerable [23] to $\epsilon$-bound adversarial attacks such as FGSM [23], PGD [42], CW [11], and Momentum [17]; Unrestricted attacks [7, 19] can significantly degrade model performance as well. Defense strategies based on heuristics such as feature squeezing [82], denoising [80], encoding [10], specialized nonlinearities [83] and distillation [56] have had limited success against stronger attacks [2]. On the other hand, standard adversarial training [42] and its variants such as [85] are fairly effective on datasets such as `MNIST`, `CIFAR-10` and `CIFAR-100`. However, on larger datasets such as `ImageNet`, these methods have limited success [63]; recent attempts [78, 63] that make adversarial training faster do not improve robustness either. In Appendix E, we show that $\ell_2$ adversarial training on synthetic datasets can improve robustness by some extent but it is unable to learn optimal large-margin $\ell_2$-robust classifiers.

**Detecting OOD Examples**: Neural networks trained using standard training procedures tend to rely on low-level features and spurious correlations and hence exhibit brittleness to benign distributional changes to the data. Recent works thus aim to detect OOD examples using generative models [57], statistical tests [59], and model confidence scores [29, 40, 39]. Our experiments in Section 4 that validate extreme SB in practice also show that detectors that directly or indirectly rely on model scores to detect OOD examples may not work well as SGD-trained neural networks can exhibit complete invariance to predictive-but-complex features.

# B  Experiment Details

In this section, we provide additional details on the datasets, models, optimization methods and training hyperparameters used in our experiments.

**One-dimensional Building Blocks**: We first describe the data generation process underlying each building block: linear, noisy linear, and $k$-slab. Then, we introduce a noisy version of the 5-slab block, which we later use in Appendix D.

- Linear$(\gamma, B)$: The linear block is parameterized by the effective margin $\gamma$ and width $B$. The distribution first samples a label $y \in \{-1, 1\}$ uniformly at random, and then given $y$, $x$ is sampled as follows: $x = y(B\gamma + (B - B\gamma) \cdot U(0, 1))$, where $U(0, 1)$ is the uniform distribution on $[0, 1]$.

- NoisyLinear$(\gamma, B, p)$: The noisy linear block is parameterized by effective margin $\gamma$, width $B$, and noise parameter $p$. Linear classifiers can attain the optimal classification accuracy of $1 - p/2$. Given label $y \in \{-1, 1\}$ sampled uniformly at random, $x$ is sampled as follows:

$$x = \begin{cases} y(B\gamma + (B - B\gamma) \cdot U(0, 1)) & \text{w.p. } p \\ U(-\gamma, \gamma) & \text{w.p. } 1-p \end{cases}$$

- Slab$(\gamma, B, k)$: The $k$-slab block is parameterized by effective margin $\gamma$, width $B$, and number of slabs $k$. We use $k \in \{3, 5, 7\}$ in our paper. The width of each slab, $w_k = 2B(1 - (k-1)\gamma)/k$, in the $k$-slab block is chosen such that the farthest points are at $-B$ and $B$. For example, given label $y \in \{-1, 1\}$ and random sign $z \in \{-1, 1\}$ sampled unif. at random, we can sample $x$ from a 3-slab block as follows:

$$x = \begin{cases} z(\frac{1}{2}w_3 \cdot U(0, 1)) & \text{if } y = -1 \\ z(\frac{1}{2}w_3 + 2B\gamma + w_3 \cdot U(0, 1)) & \text{if } y = +1 \end{cases}$$

For $k$-slab blocks with $k \in \{5, 7\}$, the probability of sampling from the two slabs (one on each side) that are farthest away from the origin are $1/4$ and $1/8$ respectively to ensure that the variance of instances in positive and negative classes, $x_+$ and $x_-$, are equal.

- NoisySlab$(\gamma, B, k, p)$: Analogous to the noisy linear block, the noisy variant of the $k$-slab block is additionally parameterized by a noise parameter $p$. In this setting, a $(k-1)$-piecewise linear classifier can attain the optimal classification accuracy of $1 - p/2$. For example, For example, given label $y \in \{-1, 1\}$ and random sign $z \in \{-1, 1\}$ sampled uniformly at random, we can sample $x$ from a $p$-noisy 3-slab block as follows:

$$x = \begin{cases} \begin{cases} z(\frac{1}{2}w_3 \cdot U(0, 1)) & \text{if } y = -1 \\ z(\frac{1}{2}w_3 + 2B\gamma + w_3 \cdot U(0, 1)) & \text{if } y = +1 \end{cases} & \text{w.p. } 1 - p \\ z(\frac{1}{2}w_3 + (2B\gamma - \frac{1}{2}w_3) \cdot U(0, 1)) & \text{w.p. } p \end{cases}$$

**Datasets**: We now outline the default hyperparameters for generating the synthetic datasets used in the paper, provide additional details on the `LSN` dataset, and introduce two additional synthetic datasets as well as multiple versions of the `MNIST-CIFAR` dataset (i.e., with different class pairs).

- Synthetic Dataset Hyperparameters: Recall that we use four $d$-dimensional synthetic datasets— `LMS-k`, `L̂MS-k`, `MS-(5,7)`, and `MS-5`—wherein each coordinate corresponds to one of the building blocks described above. Unless mentioned otherwise, for all four datasets, we set the effective margin parameter $\gamma = 0.1$, width parameter $B = 1$, and noise parameter $p = 0.1$ in all blocks/coordinates. Also recall that each dataset comprises at most one "simple" feature `S` and multiple independent complex features $S^c$. In our experiments, all datasets have sample sizes that are large enough for all models considered in the paper to learn complex features $S^c$ and attain optimal test accuracy, even in the absence of `S`; we use sample sizes of $50000$ for `LMS-5` and `MS-5` and $40000$ for `L̂MS-7`.

- LSN Dataset: Recall that the `LSN` dataset (described in Section 3) is a stylized version of the `LMS-k` that is amenable to theoretical analysis. In `LSN`, conditioned on the label $y$, the first and second coordinates of $x$ are *singleton* linear and 3-slab blocks: linear and 3-slab blocks have support on

$\{-1, 1\}$ and $\{-1, 0, 1\}$ respectively. The remaining coordinates are standard gaussians and not predictive of the label. Each data point $(x_i, y_i) \in \Re^d \times \{-1, 1\}$ can be sampled as follows:

$$y_i = \pm 1, \text{ w.p. } 1/2, \quad \varepsilon_i = \pm 1, \text{ w.p. } 1/2,$$
$$x_{i1} = y_i \qquad\qquad\qquad\qquad\qquad \text{(Linear coordinate)},$$
$$x_{i2} = \left(\frac{y_i + 1}{2}\right)\varepsilon_i \qquad\qquad\qquad \text{(Slab coordinate)},$$
$$x_{i,3:d} \sim \mathcal{N}(0, I_{d-2}) \qquad\qquad\quad (d{-}2 \text{ Noise coordinates)}.$$

- Additional Datasets: We now introduce $\hat{\texttt{MS}}$-`(5,7)`, the noisy version of `MS-(5,7)`, and three `MNIST-CIFAR` datasets, each with different `MNIST` and `CIFAR10` classes.

  - $\hat{\texttt{MS}}$-`(5,7)`: Noisy 5-slab and multiple noiseless 7-slab blocks; the first coordinate is a noisy 5-slab block and the remaining $d{-}1$ coordinates are independent 7-slab blocks. Note that this dataset comprises a noisy-but-simpler 5-slab block and multiple noiseless 7-slab blocks; a 6-piecewise linear classifier can attain 100% accuracy by learn any 7-slab block.

  - `MNIST-CIFAR` datasets: Recall that images in the `MNIST-CIFAR` datasets are concatenations of `MNIST` and `CIFAR10` images. We introduce additional variants of the `MNIST-CIFAR` using different class pairs to show that our results in the paper are robust to the exact choice of pairs:

| Datasets | Class −1 | | Class +1 | |
|---|---|---|---|---|
| | MNIST | CIFAR10 | MNIST | CIFAR10 |
| MNIST-CIFAR:A | Digit 0 | Automobile | Digit 1 | Truck |
| MNIST-CIFAR:B | Digit 1 | Automobile | Digit 4 | Truck |
| MNIST-CIFAR:C | Digit 0 | Airplane | Digit 1 | Ship |

Table 3: Three `MNIST-CIFAR` datasets. We use `MNIST-CIFAR:A` in the paper. In `MNIST-CIFAR:B`, we use different `MNIST` classes: digits 1 and 4. In `MNIST-CIFAR:C`, we use different `CIFAR10` classes: airplane and ship. Our results in Section 4 hold on all three `MNIST-CIFAR` datasets.

**Models**: Here, we briefly describe the models (and its abbreviations) used in the paper. We use fully-connected (`FCN`s), convolutional (`CNN`s), and sequential neural networks (`GRU`s [15]) on synthetic datasets. Abbreviations $(w, d)$-`FCN` denotes `FCN` with width $w$ and depth $d$, $(f, k, d)$-`CNN` denotes $d$-layer `CNN`s with $f$ filters of size $k \times k$ in each layer with and $(h, l, d)$-`GRU` denotes $d$-layer $d$-layer `GRU` with input dimensionality $l$ and hidden state dimensionality $h$. On `MNIST-CIFAR`, we train MobileNetV2 [60], GoogLeNet [70], ResNet50 [27] and DenseNet121 [32].

**Training Procedures**: Unless mentioned otherwise, we use the following hyperparameters for standard training and adversarial training on synthetic and `MNIST-CIFAR` data:

- Standard Training: On synthetic datasets, we use Stochastic Gradient Descent (SGD) with (fixed) learning rate $0.1$ and batch size $256$, and $\ell_2$ regularization $5 \cdot 10^{-7}$. On `MNIST-CIFAR` datasets, we use SGD with initial learning rate $0.05$ with decay factor of $0.2$ every 30 epochs, momentum $0.9$ and $\ell_2$ regularization $5 \cdot 10^{-5}$. We do not use data augmentation. We run all models for at most 500 epochs and stop early if the training loss goes below $10^{-2}$.

- Adversarial Training: We use the same SGD hyperparameters (as described above) on synthetic and `MNIST-CIFAR` datasets. We use Projected Gradient Descent (PGD) Adversarial Training [42] to adversarially train models. We use learning rate $0.1$ and 40 iterations to generate $\ell_2$ & $\ell_\infty$ perturbations

# C  Additional Results on the Extreme Nature of Simplicity Bias (SB)

Recall that Section 4 of the paper establishes the extreme nature of SB: *If all features have full predictive power, NNs rely exclusively on the simplest feature* $\mathtt{S}$ *and remain invariant to all complex features* $\mathtt{S}^c$—in Section 4 of the paper. Now, we further validate the extreme nature of SB across model architectures, datasets, optimizers, activation functions and regularization. We also analyze the effect of input dimensionality, number of complex features, choice and scaling of random initialization and non-random initialization.

## C.1  Effect of Model Architecture

In this section, we supplement our results in Section 4 of the paper by showing that extreme simplicity bias (SB) persists across several model architectures and on synthetic as well as image-based datasets. In Table 4, we present $\{\mathtt{S},\mathtt{S}^c\}$-Randomized AUCs for FCNs, CNNs and GRUs with depth $\{1,2\}$ trained on LMS and MS-(5,7) datasets and state-of-the-art CNNs trained on MNIST-CIFAR:A. While the $\mathtt{S}^c$-randomized AUC equals 1.00 (perfect classification), we see that the $\mathtt{S}$-randomized AUCs are approximately 0.5 for all models. This is because all models essentially only rely on the simplest feature $\mathtt{S}$ and remain invariant to all complex features $\mathtt{S}^c$, even though all features have equal predictive power.

| Dataset | Set S | Set $\mathtt{S}^c$ | Model | Randomized AUC | |
| --- | --- | --- | --- | --- | --- |
| | | | | Set S | Set $\mathtt{S}^c$ |
| LMS-5 | Linear | 5-Slabs | (100,1)-FCN | 0.50 | 1.00 |
| | | | (100,2)-FCN | 0.49 | 1.00 |
| | | | (32,7,1)-CNN | 0.50 | 1.00 |
| | | | (32,7,2)-CNN | 0.50 | 1.00 |
| | | | (100,10,1)-GRU | 0.51 | 1.00 |
| | | | (100,10,2)-GRU | 0.50 | 1.00 |
| MS-(5,7) | 5-Slab | 7-Slabs | (100,1)-FCN | 0.50 | 1.00 |
| | | | (100,1)-FCN | 0.50 | 1.00 |
| | | | (32,7,1)-CNN | 0.50 | 1.00 |
| | | | (32,7,2)-CNN | 0.50 | 1.00 |
| | | | (100,10,1)-GRU | 0.50 | 1.00 |
| | | | (100,10,2)-GRU | 0.50 | 1.00 |
| MNIST-CIFAR:A | MNIST block | CIFAR block | MobileNetV2 | 0.52 | 1.00 |
| | | | GoogLeNet | 0.51 | 1.00 |
| | | | ResNet50 | 0.50 | 1.00 |
| | | | DenseNet121 | 0.52 | 1.00 |

Table 4: Extreme SB across models trained on synthetic and image-based datasets show that all models exclusively rely on the simplest feature $\mathtt{S}$ and remain completely invariant to all complex features $\mathtt{S}^c$

## C.2  Effect of MNIST-CIFAR Class Pairs

In this section, we supplement our results on MNIST-CIFAR (in Section 4) in order to show that extreme SB observed in MobileNetV2 [60], GoogLeNet [70], ResNet50 [27] and DenseNet121 [32] does not depend on the exact choice of MNIST and CIFAR10 class pairs used to construct the MNIST-CIFAR datasets. To do so, we evaluate the MNIST-randomized and CIFAR10-randomized metrics of the aforementioned models on three datasets–MNIST-CIFAR:A, MNIST-CIFAR:B, MNIST-CIFAR:C—described in Appendix B.

Table 5 presents the standard, MNIST-randomized and CIFAR10-randomized AUC values of MobileNetV2, GoogLeNet, ResNet50 and DenseNet121 on three MNIST-CIFAR datasets. We observe that randomizing over the simpler MNIST block is sufficient to fully degrade the predictive power of all models; for instance, randomizing the MNIST block drops the AUC values of ResNet50 from 1.0 to 0.5 (i.e., equivalent to random classifier). However, randomizing the CIFAR10 block has no effect—standard AUC and CIFAR10-randomized AUCs equal 1.0. In contrast, an ideal classifier that relies on MNIST & CIFAR10 would attain non-trivial AUC even when the MNIST block is randomized.

| Model | MNIST-CIFAR:A AUCs | | | MNIST-CIFAR:B AUCs | | | MNIST-CIFAR:C AUCs | | |
|---|---|---|---|---|---|---|---|---|---|
| | Standard | CIFAR10 Randomized | MNIST Randomized | Standard | CIFAR10 Randomized | MNIST Randomized | Standard | CIFAR10 Randomized | MNIST Randomized |
| MobileNetV2 | $1.00 \pm 0.00$ | $1.00 \pm 0.00$ | $0.53 \pm 0.01$ | $1.00 \pm 0.00$ | $1.00 \pm 0.00$ | $0.53 \pm 0.02$ | $1.00 \pm 0.00$ | $1.00 \pm 0.00$ | $0.50 \pm 0.01$ |
| GoogLeNet | $1.00 \pm 0.00$ | $1.00 \pm 0.00$ | $0.52 \pm 0.02$ | $1.00 \pm 0.00$ | $1.00 \pm 0.00$ | $0.50 \pm 0.01$ | $1.00 \pm 0.00$ | $1.00 \pm 0.00$ | $0.53 \pm 0.01$ |
| ResNet50 | $1.00 \pm 0.00$ | $1.00 \pm 0.00$ | $0.50 \pm 0.01$ | $1.00 \pm 0.00$ | $1.00 \pm 0.00$ | $0.51 \pm 0.01$ | $1.00 \pm 0.00$ | $1.00 \pm 0.00$ | $0.50 \pm 0.03$ |
| DenseNet121 | $1.00 \pm 0.00$ | $1.00 \pm 0.00$ | $0.53 \pm 0.02$ | $1.00 \pm 0.00$ | $1.00 \pm 0.00$ | $0.52 \pm 0.01$ | $1.00 \pm 0.00$ | $1.00 \pm 0.00$ | $0.54 \pm 0.01$ |

Table 5: (Extreme SB in three `MNIST-CIFAR` datasets) Standard and randomized AUCs of four state-of-the-art CNNs trained on three `MNIST-CIFAR` datasets. The AUC values collectively indicate that all models exclusively rely on the `MNIST` block.

## C.3 Effect of Optimizers and Activation Functions

Now, we study the effect of activation function and optimizer on extreme SB. That is, can the usage of different activation functions and optimizer encourage trained neural networks to rely on complex features $S^c$ in addition to the simplest feature $S$?

| Activation Function | LMS-7 | | | MS-(5,7) | | |
|---|---|---|---|---|---|---|
| | SGD | Adam | RMSProp | SGD | Adam | RMSProp |
| ReLU | $0.499 \pm 0.001$ | $0.497 \pm 0.003$ | $0.502 \pm 0.004$ | $0.499 \pm 0.003$ | $0.499 \pm 0.004$ | $0.496 \pm 0.004$ |
| Leaky ReLU | $0.501 \pm 0.001$ | $0.497 \pm 0.003$ | $0.501 \pm 0.005$ | $0.499 \pm 0.005$ | $0.498 \pm 0.002$ | $0.498 \pm 0.005$ |
| PReLU | $0.500 \pm 0.004$ | $0.500 \pm 0.003$ | $0.501 \pm 0.004$ | $0.501 \pm 0.004$ | $0.496 \pm 0.003$ | $0.499 \pm 0.002$ |
| Tanh | $0.495 \pm 0.001$ | $0.502 \pm 0.004$ | $0.495 \pm 0.004$ | $0.498 \pm 0.004$ | $0.499 \pm 0.004$ | $0.498 \pm 0.002$ |

Table 6: (Effect of activation function and optimizers) $(100, 2)$-FCNs with multiple activation functions—ReLU, Leaky ReLU [41], PReLU [28], and Tanh—trained on `LMS-5` data using common first-order optimization methods—SGD, Adam [38], and RMSProp [72]—exhibit extreme SB.

Table 6 presents the `S`-randomized AUCs of $(100, 2)$-FCNs with multiple activation functions—ReLU, Leaky ReLU [41], PReLU [28], and Tanh—trained on `LMS-7` and `MS-(5,7)` datasets using multiple commonly-used optimizers: SGD, Adam [38],and RMSProp [72]. We observe that for all combinations of activations and optimizers, trained FCNs still only rely on simplest feature `S`; `S`-randomized and $S^c$-randomized AUCs are approximately $0.50$ and $1.0$ respectively for all optimizers and activation functions. Therefore, in addition to SGD, commonly used first-order optimization methods such as Adam and RMSProp cannot jointly learn large-margin classifiers that rely on learn slab-structured features in the presence of a noisy linear structure. To summarize, the experiment in Appendix C.2 shows that simply altering the choice of optimizer and activation function does not have any effect on extreme SB. Similar to the experiments in Section 4 of the paper, all models exclusively rely on simplest feature `S` and remain invariant to complex features $S^c$.

## C.4 Effect of $\ell_2$ Regularization and Dropout

In this section, we use SGD-trained `FCNs` trained on `LMS-7` data to examine the extent to which Dropout [68] and $\ell_2$ regularization alters the extreme nature of SB. Specifically, we use Dropout probability parameter $\{0.0, 0.05, 0.10\}$ and $\ell_2$ regularization parameters $\{0.01, 0.001\}$ when training `FCNs` with width 100 and depth $\{1, 2\}$ on `LMS-7` data using SGD. In Table 7, we show the standard and $S^c$-randomized AUCs equal $1.00$ (perfect classification), whereas the `S`-randomized AUCs are approximately $0.5$ for all models. Applying Dropout while reducing the amount of $\ell_2$ regularization has negligible effect on the extreme nature of SB observed in the synthetic or image-based datasets.

## C.5 Effect of Input Dimension and Number of Complex Features

In this section, we evaluate the performance of `FCNs` trained on `LMS-7` data using SGD to show the extreme SB persists in the low-dimensional setting ($d < 10$) and also with varying number of 7-slab features $1 \le |S^c| \le d$. As shown in Table 8, decreasing the input dimension $d$ or the number of complex features $|S^c|$ has no effect on extreme SB of FCNs trained on the `LMS-7` dataset. Similar to our results in Figure 3, the standard and randomized AUCs collectively show that the SGD-trained $(100,1)$-FCNs exclusively rely on the linear component and do not rely on the 7-slab coordinates.

(a)  (b)  (c)

Figure 5: Effect of non-random initialization on simplicity bias. Subplot (a) standard test accuracy (on MS-7 data) of linearly-interpolated model $M_\alpha \equiv \alpha \cdot M_{slab} + (1-\alpha) \cdot M_{rand}$ increases monotonically with the interpolation constant $\alpha$. Subplots (b) and (c) show how standard, S-randomized and $S^c$-randomized AUCs vary with interpolated model $M_\alpha$ before and after training on LMS-7 data.

## C.6 Effect of Random Initialization Scale

Now, we analyze the effect of the choice and scale (i.e., magnitude of the weights of randomly initialized FCNs) of random initialization on simplicity bias using FCNs trained on LMS-7 data.

As shown in Table 9, the choice (Kaiming and Xavier) and the scale of random initialization do not alter the extreme SB phenomenon on the LMS-7 dataset. That is, scaling the randomly initialized models by up to $0.1$ and $10.0$ has no effect on simplicity bias—SGD-trained (100,1)-FCNs exclusively rely on the linear component and do not rely on the 7-slab coordinates.

## C.7 Effect of Non-random Initialization

In this section, we investigate the effect of non-random initialization on simplicity bias using FCNs trained on MS-7 and LMS-7 data. The goal of this experiment is to determine the extent to which simplicity bias persists when the untrained network (at timestep $t = 0$) attains non-random standard accuracy by relying on one or more "complex" 7-slab features.

To vary the degree of non-random initialization $\alpha$, we obtain model $M_\alpha$ by linearly interpolating the weights of a randomly initialized network $M_{rand}$ and a network $M_{slab}$ that exclusively relies on one or more "complex" 7-slab features to attain $100\%$ accuracy on the LMS-7 dataset. That is, $M_\alpha \equiv \alpha \cdot M_{slab} + (1-\alpha) \cdot M_{rand}$. Note that $M_{slab}$ is trained on MS-7 data to attain $100\%$ standard accuracy by relying on one or more 7-slab coordinates. As shown in Figure 5(a), increasing the interpolation constant $\alpha$ monotonically increases the standard accuracy of $M_\alpha$ on MS-7 data.

Now, we use the linearly interpolated model $M_\alpha$ (for varying values of $\alpha$) as initialization and train $M_\alpha$ on LMS-7 data, which additionally consists of a "simple" linearly-separable coordinate. To maintain the input dimensionality, we obtain LMS-7 data by replacing a 7-slab coordinate by the simpler linear coordinate. In order to maintain the non-random accuracy of $M_\alpha$, we use coordinate-randomized AUCs to choose and replace a 7-slab coordinate that the model does not depend on.

| Model | Dropout | Standard AUC | | S-Randomized AUC | | $S^c$-Randomized AUC | |
|---|---|---|---|---|---|---|---|
| | | $\lambda = 10^{-2}$ | $\lambda = 10^{-4}$ | $\lambda = 10^{-2}$ | $\lambda = 10^{-4}$ | $\lambda = 10^{-2}$ | $\lambda = 10^{-4}$ |
| | 0.00 | $1.00 \pm 0.00$ | $1.00 \pm 0.00$ | $0.50 \pm 0.00$ | $0.50 \pm 0.01$ | $1.00 \pm 0.00$ | $1.00 \pm 0.00$ |
| (100,1)-FCN | 0.05 | $1.00 \pm 0.00$ | $1.00 \pm 0.00$ | $0.50 \pm 0.00$ | $0.50 \pm 0.00$ | $1.00 \pm 0.00$ | $1.00 \pm 0.00$ |
| | 0.10 | $1.00 \pm 0.00$ | $1.00 \pm 0.00$ | $0.50 \pm 0.00$ | $0.50 \pm 0.00$ | $1.00 \pm 0.00$ | $1.00 \pm 0.00$ |
| | 0.00 | $1.00 \pm 0.00$ | $1.00 \pm 0.00$ | $0.50 \pm 0.00$ | $0.50 \pm 0.00$ | $1.00 \pm 0.00$ | $1.00 \pm 0.00$ |
| (100,2)-FCN | 0.05 | $1.00 \pm 0.00$ | $1.00 \pm 0.00$ | $0.50 \pm 0.00$ | $0.50 \pm 0.00$ | $1.00 \pm 0.00$ | $1.00 \pm 0.00$ |
| | 0.10 | $1.00 \pm 0.00$ | $1.00 \pm 0.00$ | $0.50 \pm 0.00$ | $0.50 \pm 0.00$ | $1.00 \pm 0.00$ | $1.00 \pm 0.00$ |

Table 7: Dropout and $\ell_2$ regularization have no effect on extreme SB of FCNs trained on LMS-7 datasets. The standard and $\{S,S^c\}$-randomized AUC values of (100,1)-FCNs and (100,2)-FCNs collectively indicate that the models still exclusively latch on to S (linear block) and remain invariant to $S^c$ (7-slab blocks).

| Input Dimension $d$ | Number of 7-Slabs $\lvert S^c \rvert$ | Standard AUC | S-Randomized AUC | $S^c$-Randomized AUC |
|---|---|---|---|---|
| 2 | 1 | $1.00 \pm 0.00$ | $0.51 \pm 0.02$ | $1.00 \pm 0.00$ |
| 5 | 1 | $1.00 \pm 0.00$ | $0.50 \pm 0.00$ | $1.00 \pm 0.00$ |
|   | 3 | $1.00 \pm 0.00$ | $0.50 \pm 0.00$ | $1.00 \pm 0.00$ |
| 10 | 1 | $1.00 \pm 0.00$ | $0.50 \pm 0.00$ | $1.00 \pm 0.00$ |
|    | 3 | $1.00 \pm 0.00$ | $0.50 \pm 0.01$ | $1.00 \pm 0.00$ |
|    | 6 | $1.00 \pm 0.00$ | $0.50 \pm 0.00$ | $1.00 \pm 0.00$ |

Table 8: Effect of input dimension and number of complex features on extreme SB of FCNs trained on LMS-7 data. The standard and $\{S, S^c\}$-randomized AUCs of (100,1)-FCNs collectively indicate that the models exclusively latch on to S (linear) and remain invariant to $S^c$ (7-slabs).

| Initialization | Scaling Factor | Standard AUC | S-Randomized AUC | $S^c$-Randomized AUC |
|---|---|---|---|---|
| Kaiming | 0.1 | $1.00 \pm 0.00$ | $0.50 \pm 0.00$ | $1.00 \pm 0.00$ |
|         | 0.5 | $1.00 \pm 0.00$ | $0.49 \pm 0.01$ | $1.00 \pm 0.00$ |
|         | 2.0 | $1.00 \pm 0.00$ | $0.50 \pm 0.00$ | $1.00 \pm 0.00$ |
|         | 10.0 | $1.00 \pm 0.00$ | $0.50 \pm 0.00$ | $1.00 \pm 0.00$ |
| Xavier | 0.1 | $1.00 \pm 0.00$ | $0.50 \pm 0.00$ | $1.00 \pm 0.00$ |
|        | 0.5 | $1.00 \pm 0.00$ | $0.50 \pm 0.00$ | $1.00 \pm 0.00$ |
|        | 2.0 | $1.00 \pm 0.00$ | $0.49 \pm 0.01$ | $1.00 \pm 0.00$ |
|        | 10.0 | $1.00 \pm 0.00$ | $0.51 \pm 0.01$ | $1.00 \pm 0.00$ |

Table 9: Effect of choice (Kaiming and Xavier) and scale of random initialization on simplicity bias of FCNs trained on LMS-7 data. The AUCs of (100,1)-FCNs collectively show increasing the scale of random initialization does not alleviate extreme simplicity bias in this setting.

As expected, Figure 5(b) shows that prior to training on LMS-7 data, the interpolated models $M_\alpha$ attain $S^c$-randomized AUC 0.50 and $S^c$-randomized accuracy equals standard AUC because the models exclusively rely on one or more 7-slab coordinates. However, as shown in Figure 5(c), after training $M_\alpha$ on LMS-7 data, the models no longer exhibit exclusive reliance on 7-slab features. In particular, when $\alpha \leq 0.5$, the models now exclusively rely on the linear coordinate only. When $\alpha \geq 0.5$, the dependence on the 7-slab coordinates is considerably reduced. Surprisingly, even when $\alpha = 1.0$, we observe that the model additionally relies on the linear coordinate, possibly due to non-zero training loss at initialization. These results collectively suggest that in this setting, non-random initialization by first training the model on complex features only can be effective in mitigating extreme simplicity bias to some extent.

# D  Additional Results on the Effect of Extreme SB on Generalization

Recall that in Section 5 of the paper, we showed that extreme SB can result in suboptimal generalization of SGD-trained models on the same data distribution. In this section, we present additional information about the experiment setup used in Section 5 to show that SB can worsen standard generalization.

## D.1  Experiment Setup in Section 5

We now provide additional information about the experimental setup used in Section 5 of the paper, where we show that extreme simplicity bias can result in suboptimal generalization. We train fully-connected networks (FCNs) of width $\{100, 200, 300\}$ and depth $\{1, 2\}$ using SGD on 50-dimensional $\hat{\mathsf{L}}\mathsf{MS}\text{-}7$ dataset of $40000$ samples, which comprises of a noisy linear coordinate ($10\%$ noise) and $49$ 7-slab coordinates. For each model architecture, we perform a grid search over SGD hyperparameters—learning rate, batch size, momentum, and weight decay—and report standard and randomized test accuracies of the model (in Table 2) that perform best on a $\hat{\mathsf{L}}\mathsf{MS}\text{-}7$ validation dataset. We perform a grid search over the following SGD hyperparameters:

- Learning rate: $\{0.001, 0.01, 0.05, 0.1, 0.3\}$
- Batch size: $\{4, 16, 64, 256\}$
- Weight decay: $\{0, 0.00005, 0.0005\}$
- Momentum: $\{0, 0.9, 0.95\}$

We train all models for at most 10 million updates with constant learning rate schedule and stop early if the training loss diverges or goes below $0.01$. In this setting, 10 million updates is equivalent to $1000$ epochs with batch size $4$ and $64000$ epochs with batch size $256$. As mentioned in Section 5, the training sample size of $40000$ data points is large enough for FCNs of depth $\{1, 2\}$ and width $\{100, 200, 300\}$ trained on $\mathsf{MS}\text{-}7$ data (i.e., $\mathsf{S}^c$ *only*, after removing $\mathsf{S}$ from data) to attain $\approx 100\%$ test accuracy using SGD with *learning rate* $0.3$*, batch size* $256$*, weight decay* $0.0005$*, and momentum* $0.9$. The optimal hyperparameters for FCNs trained on $\hat{\mathsf{L}}\mathsf{MS}\text{-}7$ data are provided in Table 10. Note that we do not consider $\ell_2$ weight decay values larger than $0.0005$ because it results in $95\%$ test and train accuracy even after 10 million updates by preventing FCNs from overfitting to the noise in the linear component. Also note that (100,1)-FCNs are not able to completely overfit due to insufficient representation capacity when trained with the chosen SGD hyperparameters (see Table 10).

| Hyperparameter | (100,1)-FCN | (200,1)-FCN | (300,1)-FCN | (100,2)-FCN | (200,2)-FCN | (300,2)-FCN |
|---|---|---|---|---|---|---|
| Learning rate | 0.30 | 0.10 | 0.01 | 0.30 | 0.10 | 1.00 |
| Batch size | 16 | 256 | 16 | 64 | 64 | 256 |
| Weight decay | 0.0 | 0.0 | 0.0 | 0.0 | 0.0005 | 0.0005 |
| Momentum | 0.90 | 0.0 | 0.0 | 0.0 | 0.0 | 0.0 |

Table 10: SGD hyperparameters of SGD-trained FCNs trained on $\hat{\mathsf{L}}\mathsf{MS}\text{-}7$ data that result in the best validation accuracy out of all $180$ hyperparameter combinations listed in Appendix D.1.

# E   Can we mitigate Simplicity Bias?

In this section, we investigate whether standard approaches for improving generalization error and adversarial robustness—ensembles and adversarial training—help in mitigating SB.

## E.1   Ensemble Methods

We now study the extent to which ensembles mitigate SB and its adverse effect on generalization. Specifically, we evaluate the performance of ensembles of fully-connected networks (FCNs) that are trained on two datasets: L̂MS-7 and MS-5. Recall that the L̂MS-7 data comprises one simple-but-noisy linear coordinate and multiple relatively complex 7-slab coordinates that have no noise, whereas MS-5 data comprises multiple noiseless 5-slab coordinates only.

(a)                                   (b)

Figure 6: Ensembles improve performance on MS-5 data that comprises features with equal predictive power and simplicity. However, as shown in (a), ensembles do not improve performance on L̂MS-7 data that has a simple-but-noisy linear coordinate that has less predictive power than than the 7-slab coordinates; this is because individual FCNs trained on L̂MS-7 data exclusively rely on the noisy linear coordinate and consequently misclassify the same set of instances, as shown in subplot (b).

To better highlight the effect of ensembles on generalization, we choose a sample size (for both datasets) such that individual models (a) overfit to training data (i.e., non-zero generalization gap) but (b) still attain non-trivial test accuracy. We now discuss the performance of ensembles of independently trained models on MS-5 and L̂MS-7 datasets:

- MS-5 **data**: Recall that MS-5 data comprises multiple independent 5-slab blocks, one in each coordinate, that have equal simplicity and predictive power. Thus, since all features have equal simplicity, independent SGD-trained (100,2)-FCN end up relying on different 5-slab coordinates due to random initialization, as shown in Figure 6(b). As the training sample size is small, FCNs overfit to the training data and attain approximately $75\%$ test accuracy, as shown in Figure 6. Consequently, as shown in Figure 6, ensembles of these models rely on all 5-slab coordinates learned by the individual models and attain better test accuracy by aggregating model predictions and averaging out overfitting. For example, Figure 6 shows that ensembles of size 5 and 10 improves generalization by approximately $15\%$ and $20\%$ respectively.

- L̂MS-7 **data**: Recall that L̂MS-5 data comprises one simple-but-noisy linear block (with $50\%$ noise) and multiple independent 7-slab blocks that have no noise. Now, due to extreme SB, every independently trained FCN exclusively latches on (and overfits to) the simpler-but-noisy linear block, as shown in Figure 6(b). As a result, all models collectively lack diversity and essentially learn the same decision boundary because of extreme SB. Therefore, ensembles of these models do not improve generalization because the independent models make misclassifications on the same instances. As shown in Figure 6(a), ensembles of size 3, 5 and 10 do not improve generalization—the test accuracy remains $75\%$.

The ensemble performance on MS-5 data indicates that when datasets have *multiple* equally simple features, ensembles of independently trained models mitigate SB to some extent by aggregating predictions of models that rely on simple features. Conversely, the ensemble performance on L̂MS-7 data suggests that when datasets comprise few features that the more noisy and less predictive than the rest, ensembles may not improve generalization. Our results also suggest that the generalization improvements using ensemble methods in practice may stem from combining multiple simple-but-noisy features (such as color, texture) and not by learning complex features (such as shape).

## E.2 Adversarial Training

We now investigate the extent to which adversarial training [42] mitigates SB and its adverse effect on adversarial robustness using two datasets: MNIST-CIFAR and AdvMS-(5,7).

### E.2.1 Adversarially training FCNs on AdvMS-(5,7) data

Now, we first introduce AdvMS-(5,7), a variant of the MS-(5,7) dataset, and then investigate if adversarially trained FCNs that improve adversarial robustness by some extent also mitigate extreme simplicity bias (SB).

AdvMS-(5,7) dataset: Recall that $d$-dimensional MS-(5,7) data, introduced in Section 3, consists of $d-1$ 7-slab coordinates and a single relatively simpler 5-slab coordinate, all of which have perfect predictive power. Similar to MS-(5,7) data, the $d$-dimensional AdvMS-(5,7) data comprises 5-slab and 7-slab coordinates. Specifically, the first $d/2$ coordinates correspond to independent 5-slabs, each with effective margin $\gamma_5$ and the other $d/2$ coordinates correspond to independent 7-slabs with effective margin $\gamma_7$. In contrast to MS-(5,7) data, the AdvMS-(5,7) dataset (a) comprises $d/2$ 5-slab coordinates and (b) the 5-slabs and 7-slabs do not necessarily share the same effective margin. In our experiments below, we set $d = 20$, $\gamma_5 = 0.05$ and $\gamma_7 = 0.15$. That is, we conduct our experiments on 20-dimensional data in which the the simple features S and complex features $S^c$ correspond to the 10 small-margin 5-slab and 10 large-margin 7-slab coordinates respectively.

Experiment setup: In Section 4, we observed that SGD-trained FCNs, due to extreme SB, exclusively rely on the simplest feature S and consequently learn small-margin classifiers that are highly vulnerable to adversarial attacks. To mitigate SB and attain optimal adversarial robustness, fully-connected networks with width $w$ and depth $d$, $(w, d)$-FCNs, must learn maximum-margin classifiers and consequently rely on *all* simple and complex features (i.e., features in S and $S^c$). The margin of the 20-dimensional AdvMS-(5,7) dataset described above is approximately $\gamma_{\text{data}} = 0.62$, which implies that the maximum-margin classifier should exhibit robustness to $\ell_2$ adversarial perturbations that have norm $\epsilon < \gamma_{\text{data}}$. Therefore, to check if adversarial training mitigates extreme SB, we evaluate the robustness of adversarially trained FCNs against $\ell_2$ adversarial perturbations that have norm $\epsilon < \gamma_{\text{data}}$. In addition to $\gamma_{\text{data}}$, let $\gamma_S = 0.30$ denote the maximum margin of the classifier that exclusively relies on all 5-slab features in S.

Also note that (a) adversarial perturbations are generated using PGD attacks [42], (b) $(200, 2)$-FCNs and $(1000, 2)$-FCNs are expressive enough to learn the maximum-margin classifier on the 20-dimensional AdvMS-(5,7) data, (c) FCNs are adversarially trained for 4000 epochs with initial learning rate 0.1 that decays by a multiplicative factor of 0.1 after every 1000 epochs, and (d) the training data comprises 6000 data points, which is enough for SGD-trained $(1000, 2)$-FCNs to learn 7-slab coordinates and attain 100% generalization.

Experiment results: Recall that the feature sets S and $S^c$ correspond to the set of ten 5-slab and 7-slab coordinates in the AdvMS-(5,7) dataset respectively. As shown in Figure 7, we evaluate the standard (blue), $\epsilon$-robust (orange), S-randomized (green) and $S^c$-randomized (red) accuracies, defined in Section 3, of FCNs with width $\{200, 1000\}$ and depth 2 that are adversarially trained with $\ell_2$ perturbation norm $\epsilon \leq \gamma_{\text{data}}$. The dashed and solid purple vertical bars denote $\gamma_S$ and $\gamma_{\text{data}}$ respectively. We make two key observations:

- Adversarial trained FCNs do not learn maximum-margin classifiers. When $\epsilon \leq 0.1$, $(200, 2)$-FCNs learn classifiers that attain 100% standard and $\epsilon$-robust accuracies. However, when $\epsilon \geq 0.2$, due

Figure 7: Adversarially trained FCNs on AdvMS-(5,7) data exhibit adversarial robustness to some extent, but are (a) unable to mitigate extreme simplicity bias, as shown by the $\{S, S^c\}$-randomized accuracies and (b) do not learn maximum-margin classifiers that attain optimal adversarial robustness (i.e., 100% $\gamma_{\text{data}}$-robust accuracy). See Appendix E.2 for more detail.

to optimization-related issues, adversarially trained $(200, 2)$-FCNs are unable to learn a non-trivial classifier that obtains more than $50\%$ standard and $\epsilon$-robust accuracy. Increasing the model width from 200 to 1000 improves adversarial robustness to some extent—adversarially trained $(1000, 2)$-FCNs learn classifiers with $100\%$ standard and robust accuracies when $\epsilon \leq 0.25$. However, when $\epsilon \geq \gamma_{\mathsf{S}} = 0.3$ (dashed purple line), adversarially trained $(2000, 2)$-FCNs are unable to learn $\epsilon$-robust classifiers as well. We note that further increasing the model width to 2000 does not improve robustness. Consequently, adversarial training does not result in maximum-margin classifiers that have optimal adversarial robustness (i.e., classifiers with $100\%$ $\gamma_{\mathrm{data}}$-robust accuracy) on `AdvMS-(5,7)` data. These results reconcile two phenomena observed in practice: larger capacity models can improve adversarial robustness [81], but large-epsilon adversarial training can "fail" and result in trivial classifiers due to optimization-related issues [66].

- Adversarial training does not mitigate extreme SB. The $\{\mathsf{S},\mathsf{S}^c\}$-randomized accuracies of adversarially trained FCNs collectively show that adversarial training does not mitigate extreme SB. When the perturbation budget $\epsilon \leq \gamma_{\mathsf{S}}$, adversarially trained $(2000, 2)$-FCNs exhibit robustness by *exclusively* relying on *multiple* "simple" 5-slab coordinates. That is, randomizing $\mathsf{S}$ drops the model accuracy to $50\%$, but randomizing the more complex 7-slab coordinates has no effect on model accuracy. Conversely, when $\epsilon \geq \gamma_{\mathsf{S}}$, classifiers with $100\%$ $\epsilon$-robust accuracy must rely on features in $\mathsf{S}$ *and* $\mathsf{S}^c$. However, as shown in Figure 7, when $\epsilon \geq \gamma_{\mathsf{S}}$, adversarial training fails and results in trivial classifiers that attain $50\%$ standard and robust accuracy.

To summarize, the experiment above shows that adversarially trained FCNs do not mitigate simplicity bias or learn maximum-margin classifiers that are optimally robust to $\ell_2$ adversarial attacks.

### E.2.2 Adversarially training `CNNs` on `MNIST-CIFAR` data

Recall that the `MNIST-CIFAR` images, described in Section 3, are vertically concatenations of "simple" `MNIST` images and the more complex `CIFAR10` images. Next, we show that models that are $\ell_{\infty}$-adversarially trained on the `MNIST-CIFAR` data improve $\epsilon$-robust accuracy (defined in Section 3) but do not achieve the best possible $\epsilon$-robust accuracy due to extreme SB.

Table 11 evaluates the standard, $\epsilon$-robust and `CIFAR10`-randomized accuracies of SGD-trained and adversarially trained MobileNetV2, ResNet50 and DenseNet121 using the `MNIST-CIFAR` dataset. First, we observe that adversarial training with perturbation norm 0.3 significantly improves $\epsilon$-robust accuracies over those of SGD-trained models, without degrading the models' standard test accuracies. However, the `CIFAR10`-randomized accuracies indicate that the adversarial training does not lead to reliance on the `CIFAR10` block—adversarially trained `CNNs` continue to remain invariant to the `CIFAR10` block even though it is almost fully predictive of the label. Consequently, these results suggest that adversarial training improves robustness but does not achieve the best $\epsilon$-robust accuracy on the `MNIST-CIFAR` dataset.

To summarize, our experiments on `AdvMS-(5,7)` and `MNIST-CIFAR` datasets show that while adversarial training does improve the $\epsilon$-robust accuracy over that of SGD-trained model, adversarially trained models continue to remain susceptible to extreme SB and consequently do not achieve maximum possible adversarial robustness.

| Model | $\ell_{\infty}$ budget $\varepsilon$ | Test Accuracy | | $\epsilon$-Robust Accuracy | | CIFAR10-Randomized Accuracy | |
|---|---|---|---|---|---|---|---|
| | | Standard SGD | $\ell_{\infty}$ Adv. Training | Standard SGD | $\ell_{\infty}$ Adv. Training | Standard SGD | $\ell_{\infty}$ Adv. Training |
| MobileNetV2 | 0.30 | $0.999 \pm 0.001$ | $0.999 \pm 0.000$ | $0.000 \pm 0.000$ | $0.991 \pm 0.000$ | $0.493 \pm 0.005$ | $0.493 \pm 0.001$ |
| DenseNet121 | 0.30 | $1.000 \pm 0.000$ | $0.999 \pm 0.000$ | $0.000 \pm 0.000$ | $0.981 \pm 0.003$ | $0.494 \pm 0.005$ | $0.501 \pm 0.003$ |
| ResNet50 | 0.30 | $1.000 \pm 0.000$ | $0.999 \pm 0.001$ | $0.001 \pm 0.000$ | $0.982 \pm 0.002$ | $0.501 \pm 0.001$ | $0.499 \pm 0.002$ |

Table 11: Adversarial training on `MNIST-CIFAR`: The table above presents standard, $\epsilon$-robust and `CIFAR10`-randomized accuracies of SGD-trained and adversarially trained MobileNetV2, DenseNet121 and ResNet50 models. While adversarial training significantly improves $\varepsilon$-robust accuracy, it does not encourage models to learn complex features (`CIFAR10` block in this case). The `CIFAR10`-randomized accuracies indicate that adversarially trained models do not mitigate extreme SB, as they exclusively rely on the `MNIST` block.

# F    Proof of Theorem 1

In this section, we first re-introduce the data distribution and theorem. Then, we describe the proof sketch and notation, before moving on to the proof.

**Linear-Slab-Noise (LSN) data**: The LSN dataset is a stylized version of LMS-k that is amenable to theoretical analysis. In LSN, conditioned on the label $y$, the first and second coordinates of $x$ are *singleton* linear and 3-slab blocks: linear and 3-slab blocks have support on $\{-1, 1\}$ and $\{-1, 0, 1\}$ respectively. The remaining coordinates are standard gaussians and not predictive of the label. Each data point $(x_i, y_i) \in \Re^d \times \{-1, 1\}$ from LSN can be sampled as follows:

$$y_i = \pm 1, \ \text{w.p. } 1/2, \quad \varepsilon_i = \pm 1, \ \text{w.p. } 1/2,$$

$$x_{i1} = y_i \qquad\qquad\qquad\qquad \text{(Linear coordinate)},$$

$$x_{i2} = \left(\frac{y_i + 1}{2}\right) \varepsilon_i \qquad\qquad \text{(Slab coordinate)},$$

$$x_{i,3:d} \sim \mathcal{N}(0, I_{d-2}) \qquad\qquad (d{-}2 \text{ Noise coordinates}).$$

According to Theorem 1 (re-stated), one-hidden-layer ReLU neural networks trained with standard mini-batch gradient descent (GD) on the LSN dataset provably learns a classifier that *exclusively* relies on the "simple" linear coordinate, thus exhibiting simplicity bias at the cost of margin.

**Theorem 1.** *Let $f(x) = \sum_{j=1}^{k} v_j \cdot ReLU(\sum_{i=1}^{d} w_{i,j} x_i)$ denote a one-hidden-layer neural network with $k$ hidden units and ReLU activations. Set $v_j = \pm 1/\sqrt{k}$ w.p. $1/2 \ \forall j \in [k]$. Let $\{(x^i, y^i)\}_{i=1}^{m}$ denote i.i.d. samples from LSN where $m \in [cd^2, d^\alpha/c]$ for some $\alpha > 2$. Then, given $d > \Omega(\sqrt{k} \log k)$ and initial $w_{ij} \sim \mathcal{N}(0, \frac{1}{dk \log^4 d})$, after $O(1)$ iterations, mini-batch gradient descent (over $w$) with hinge loss, step size $\eta = \Omega(\log d)^{-1/2}$, mini-batch size $\Theta(m)$, satisfies:*

- *Test error is at most $1/poly(d)$*
- *The learned weights of hidden units $w_{ij}$ satisfy:*

$$\underbrace{|w_{1j}| = \frac{2}{\sqrt{k}} \left(1 - \frac{c}{\sqrt{\log d}}\right) + O\left(\frac{1}{\sqrt{dk} \log d}\right)}_{\textit{Linear Coordinate}}, \ \underbrace{|w_{2,j}| = O\left(\frac{1}{\sqrt{dk} \log d}\right)}_{\textit{3-Slab Coordinate}}, \ \underbrace{\|w_{3:d,j}\| = O\left(\frac{1}{\sqrt{k} \log d}\right)}_{d{-}2 \ \textit{Noise Coordinates}}$$

*with probability greater than $1 - \frac{1}{poly(d)}$. Note that $c$ is a universal constant.*

**Proof Sketch** Since the number of iterations $t = O(1)$, we partition the dataset into $t$ minibatches each of size $n := m/t$ samples. This means that each iteration uses a fresh batch of $n$ samples and the $t$ iterations together form a single pass over the data. The overall outline of the proof is as follows. If the step size is $\eta$, then for $t \lesssim \frac{4}{\eta}$ iterations, with probability $\geq 1 - \frac{1}{poly(d)}$,

- Lemma 2 shows that the hinge loss is "active" (i.e., $yf(x) < 1$) for all data points in a given batch.

- Under this condition, we derive closed-form expressions for *population* gradients in Lemmas 4, 5 and 6.

- Lemma 1 uses the above lemmas to establish precise estimates of the linear, slab and noise coordinates for all iterations until $t$.

The proof is organized as follows. Appendix F.1 presents the main lemmas that will directly lead to Theorem 1. Appendix F.2 derives closed form expressions for population gradients and Appendix F.3 presents auxiliary lemmas that are useful in the main proofs.

**Notation** Recall that $f(x) = \sum_{j=1}^{k} v_j \cdot \text{ReLU}(\sum_{i=1}^{d} w_{ij} x_i) = v^T \text{ReLU}(W^T x)$ where $W \in \mathbb{R}^{d \times k}$ and $v \in \mathbb{R}^k$. Note that $w_i = [w_{1i} \, w_{2i}, \cdots w_{di}]^T$ is the $i^{th}$ column in $W$. Let $\bar{w}_i$ and $\bar{x}_j$ denote the $w_{3:d,i}$ and $x_{3:d,j}$ respectively. Also, let $\mathcal{S}_n = \{(x_i, y_i)\}_{i=1}^{n}$ denote a set of $n$ i.i.d. points randomly sampled from LSN. For simplicity, we also assume $|\{i : v_i = 1/\sqrt{k}\}| = |\{i : v_i = -1/\sqrt{k}\}| = k/2$. We can now define the loss function as $\mathcal{L}_f(\mathcal{S}_n) = 1/n \sum_i^n \ell(x_i, y_i)$, where $\ell(x, y) = \max(0, 1 - yf(x))$ denotes the hinge loss. For notational simplicity, we use $X = \mu \pm \delta$ and $|X - \mu| \leq \delta$ interchangeably. Also let $\varphi$ and $\phi$ denote the probability density function and cumulative distribution function of standard normal distribution.

*Proof of Theorem 1.* The proof directly follows from Lemma 1 and Lemma 2. In Lemma 1, we show that the weights in the linear coordinate are $\Omega(\sqrt{d})$ larger than the weights in the slab and noise coordinates. Applying Lemma 1 at $\hat{t} = \lfloor \frac{4}{\eta}(1 - \frac{c_n}{\sqrt{\log d}}) \rfloor$ gives the following result:

$$w_{1i}^{(\hat{t})} \overset{(a)}{=} \frac{2}{\sqrt{k}}(1 - \frac{c_n}{\sqrt{\log d}}) + O(\frac{1}{\sqrt{dk}\log d}) \text{ and } |w_{2i}^{(\hat{t})}| \overset{(a)}{=} O(\frac{1}{\sqrt{dk}\log d}) \text{ and } ||\bar{w}_i^{(\hat{t})}| \overset{(a)}{=} O(\frac{1}{\sqrt{k}\log d})$$

where $(a)$ is due to $c_0(1+\hat{c})^t \leq c_0 e^{\hat{c}t} \leq c_0 e^1 = O(1)$.

The $0-1$ error of the function $f$ at timestep $\hat{t}$ is small as well, because we can directly use Lemma 2 to get $\Pr(yf(x) < 0) = 2/c^3 d^6$. Therefore, the $0-1$ error is at most $\frac{2}{c^3 d^6} = O(\frac{1}{d^6})$. ∎

## F.1 Proof by Induction

In this section, we use proof by induction to show that for the first $t = O(1/\eta)$ steps, (1) the hinge loss is "active" for all data points (Lemma 2) and (2) hidden layer weights in the linear coordinate are $\Omega(\sqrt{d})$ larger than the hidden layer weights in the slab and noise coordinates (Lemma 1).

**Lemma 1.** *Let $|\mathcal{S}_n| \in [cd^2, d^\alpha/c]$ and initialization $w_{ij} \sim \mathcal{N}(0, 1/dk \log^2 d)$. Also let $\hat{c} = \eta/4$, $c_0 = 2$ and $c_n = 5\sqrt{\alpha}c_0(1+\hat{c})^t$. Then, for all $t \leq \frac{4}{\eta}(1 - c_n/\sqrt{\log d})$, $d \geq \exp((8c_n/\eta)^2)$, $\sqrt{d/\log^3(d)} > 24\sqrt{k}/c_0 c$ and $i \in [k]$, w.p. greater than $1 - O(\frac{1}{d^2})$, we have:*

$$y_i f(x_i) \leq 1 \ \forall (x_i, y_i) \in \mathcal{S}_n \tag{1}$$

$$w_{1i}^{(t)} = \frac{t\eta v_i}{2} \pm \frac{c_0(1+\hat{c})^t}{\sqrt{dk}\log d} \tag{2}$$

$$|w_{2i}^{(t)}| \leq \frac{c_0(1+\hat{c})^t}{\sqrt{dk}\log d} \tag{3}$$

$$||\bar{w}^{(t)}||_2 \leq \frac{c_0(1+\hat{c})^t}{\sqrt{k}\log d} \tag{4}$$

*Proof.* First, we prove that equations (2), (3) & (4) hold at initialization (i.e., $t = 0$) with high probability. Using 7 and 1:

$$\max_{i \in \{1,2\}} \max_{j \leq k} |w_{ij}| \leq \frac{2}{\sqrt{dk}\log d} \text{ and } \max_{i \leq k} ||\bar{w}_i|| \leq \frac{2}{\sqrt{k}\log d} \quad \text{w.p. } 1 - \frac{2}{d^4}$$

Therefore, $w_{1i} = \frac{(0)\eta v_i}{2} \pm \frac{c_0(1+\hat{c})^0}{\sqrt{dk}\log d}$ and $|w_{2i}| \leq \frac{c_0(1+\hat{c})^0}{\sqrt{dk}\log d}$ and $||\bar{w}_i|| \leq \frac{c_0(1+\hat{c})^0}{\sqrt{k}\log d}$. Since equations (2), (3) & (4) hold at $t = 0$, we can use Lemma 2 to show that the hinge loss is "active" with high probability:

$$y_i f(x_i) = \pm \frac{c_n}{\sqrt{\log d}} < 1 \quad \text{when } d \geq \exp(c_n^2)$$

Now, we assume that the inductive hypothesis—equations (1), (2), (3) and (4)—is true after every timestep $\tau$ where $\tau \in \{0, \cdots, t\}$.

We now prove that the inductive hypothesis is true at timestep $t+1$, after applying gradient descent using the $(t+1)^{th}$ batch. Since z(1) holds at timestep $t$, we can use the closed-form expression of the gradient along the linear coordinate (lemma 4) to prove that equation (2) holds at timestep $t+1$

as well:

$$w_{1i}^{(t+1)} = w_{1i}^{(t)} + \frac{\eta v_i}{4}\left[2 + \phi\left(\frac{w_{1i}+w_{2i}}{||\bar{w}_i||}\right) + \phi\left(\frac{w_{1i}-w_{2i}}{||\bar{w}_i||}\right) - 2\phi\left(\frac{w_{1i}}{||\bar{w}_i||}\right)\right] \pm \frac{5\eta v_i}{d}\sqrt{\frac{\log(cd^2)}{c}}$$

$$= w_{1i}^{(t)} + \frac{\eta v_i}{2} + \frac{\eta v_i}{2}|w_{2i}^{(t)}| \cdot \max_{|\delta|\le|w_{2i}^{(t)}|}\frac{1}{||\bar{w}^t||}\varphi\left(\frac{w_{1i}^{(t)}+\delta}{||\bar{w}_i^{(t)}||}\right) \pm \frac{5\eta v_i}{d}\sqrt{\frac{\log(cd^2)}{c}}$$

$$\overset{(a)}{=} \frac{t\eta v_i}{2} \pm \frac{c_0(1+\hat{c})^t}{\sqrt{dk}\log d} + \frac{\eta v_i}{2} + \frac{\eta v_i}{2}\frac{c_0(1+\hat{c})^t}{\sqrt{dk}\log d} \pm \frac{5\eta v_i}{d}\sqrt{\frac{\log(cd^2)}{c}}$$

$$\overset{(b)}{=} \frac{(t+1)\eta v_i}{2} \pm \frac{c_0(1+\hat{c})^t}{\sqrt{dk}\log d} \pm \eta v_i\frac{c_0(1+\hat{c})^t}{\sqrt{dk}\log d} = \frac{(t+1)\eta v_i}{2} \pm \frac{c_0(1+\hat{c})^t(1+\eta v_i)}{\sqrt{dk}\log d}$$

$$\overset{(c)}{=} \frac{(t+1)\eta v_i}{2} \pm \frac{c_0(1+\hat{c})^{t+1}}{\sqrt{dk}\log d}$$

where $(a)$ is via equation (12) in Lemma 3, $(b)$ is because $d/\log^3(d) \ge 20/c_0 e^1\sqrt{c}$ and $(c)$ is due to $\eta v_i \le \hat{c}$.

Similarly, since equation (1) holds at timestep $t$ (via the inductive hypothesis), we can use the closed-form expression of the gradient along the slab coordinate (lemma 5) to show that the weights in the slab (i.e., second) coordinate are small (equation (3)) at timestep $t+1$ as well:

$$w_{2i}^{(t+1)} = w_{2i}^{(t)} + \frac{\eta v_i}{4}\left[\phi\left(\frac{w_{1i}+w_{2i}}{||\bar{w}_i||}\right) - \phi\left(\frac{w_{1i}-w_{2i}}{||\bar{w}_i||}\right)\right] \pm \frac{5\eta v_i}{d}\sqrt{\frac{\log(cd^2)}{c}}$$

$$= w_{2i}^{(t)} + \frac{\eta v_i}{2}|w_{2i}^{(t)}| \cdot \max_{|\delta|\le|w_{2i}^{(t)}|}\frac{1}{||\bar{w}^t||}\varphi\left(\frac{w_{1i}^{(t)}+\delta}{||\bar{w}_i^{(t)}||}\right) \pm \frac{5\eta v_i}{d}\sqrt{\frac{\log(cd^2)}{c}}$$

$$\overset{(a)}{=} \pm\frac{c_0(1+\hat{c})^t}{\sqrt{dk}\log d} \pm \frac{\eta v_i}{2}\frac{c_0(1+\hat{c})^t}{\sqrt{dk}\log d} \pm \frac{\eta v_i}{2}\frac{c_0(1+\hat{c})^t}{\sqrt{dk}\log d} \le \pm\frac{c_0(1+\hat{c})^{t+1}}{\sqrt{dk}\log d}$$

where $(a)$ is due to equations (11) in Lemma 3, (3) and $d/\log^3(d) \ge 20/c_0 e^1\sqrt{c}$.

Finally, we can use the closed-form expression of the gradient along the noise coordinate (lemma 6) to prove that the norm of the gradient along the noise coordinates (i.e., coordinates 3 to $d$) is small (equation (4)) at timestep $t+1$:

$$\bar{w}_i^{(t+1)} = \bar{w}_i^{(t+1)} + \underbrace{\frac{\eta v_i}{4}\left[\varphi\left(\frac{w_{1i}+w_{2i}}{||\bar{w}_i||}\right) + \varphi\left(\frac{w_{1i}-w_{2i}}{||\bar{w}_i||}\right) - 2\varphi\left(\frac{w_{1i}}{||\bar{w}_i||}\right)\right]\frac{\bar{w}_i^{(t)}}{||\bar{w}_i||}}_{\bar{\mathcal{G}}_1} \pm \underbrace{\frac{3\eta|v_i|\log(\sqrt{cd})}{\sqrt{cd}}\frac{\bar{w}_i}{||\bar{w}_i||} \pm \frac{6\eta|v_i|}{\sqrt{cd}}u_i^{\perp}}_{\bar{\mathcal{G}}_2}$$

We first show that the the first part of the noise gradient, $\bar{\mathcal{G}}_1$, is at most $\eta v_i/2$:

$$\frac{\eta v_i}{4}\left[\varphi\left(\frac{w_{1i}+w_{2i}}{||\bar{w}_i||}\right) + \varphi\left(\frac{w_{1i}-w_{2i}}{||\bar{w}_i||}\right) - 2\varphi\left(\frac{w_{1i}}{||\bar{w}_i||}\right)\right]$$

$$= \frac{\eta v_i}{4}\underbrace{\frac{1}{||\bar{w}_i^{(t)}||}\varphi\left(\frac{w_{1i}^{(t)}}{||\bar{w}_i^{(t)}||}\right)}_{\le 1 \text{ (see eq. 13 in Lemma 3)}}\underbrace{\left[2 - \varphi\left(\frac{w_{2i}^{(t)}}{||\bar{w}_i^{(t)}||}\right)\left(\exp(\frac{w_{1i}^{(t)}w_{2i}^{(t)}}{||\bar{w}_i^{(t)}||^2}) + \exp(\frac{-w_{1i}^{(t)}w_{2i}^{(t)}}{||\bar{w}_i^{(t)}||^2})\right)\right]}_{\le 2} \le \frac{\eta v_i}{2}$$

Next, we show that the $\ell_2$ norm of the second part of the noise gradient, $||\bar{\mathcal{G}}_2||$, is $O(1/\sqrt{d})$:

$$||B|| \le 3\eta|v_i|\frac{\log(\sqrt{cd})}{\sqrt{cd}} + \frac{6\eta|v_i|}{\sqrt{cd}} \le \frac{12\eta|v_i|}{\sqrt{cd}}$$

Now, we can use the upper bounds on $\mathcal{G}_1$ and $\mathcal{G}_2$ to show that the $\ell_2$ norm of the gradient along the noise gradients is small as well:

$$||\bar{w}_i^{(t+1)}|| \le ||\bar{w}_i^{(t)}|| + \frac{\eta v_i}{2}||\bar{w}_i^{(t)}|| + \frac{12\eta|v_i|}{\sqrt{cd}} \overset{(a)}{\le} ||\bar{w}_i^{(t)}|| + \frac{\eta v_i}{2}||\bar{w}_i^{(t)}|| + \frac{\eta v_i}{2}||\bar{w}_i^{(t)}||$$

$$\overset{(b)}{\le} \frac{c_0(1+\hat{c})^t(1+\eta v_i)}{\sqrt{k}\log d} \overset{(c)}{\le} \frac{c_0(1+\hat{c})^{t+1}}{\sqrt{k}\log d}$$

where $(a)$ is because $d/\log d \ge (24\sqrt{k}/c_0 c)^2$, $(b)$ is due to equation (4) and $(c)$ is because $\eta v_i \le \hat{c}$. $\blacksquare$

Since equations (2), (3) & (4) hold at timestep $t$ (from Lemma 1), we can show that the hinge loss is positive (i.e., $yf(x) < 1$) for all data points with high probability as well.

**Lemma 2.** *Let $\mathcal{S}_n$ denote a set of $n \in [cd^2, d^\alpha/c]$ i.i.d. samples from LSN, where $\alpha > 2$ and $c > 1$. Suppose equations (2), (3) & (4) hold at timestep $t$. Also let $d \geq \exp((\frac{8c_n}{\eta})^2)$ where $c_n = 5\sqrt{\alpha}c_0(1+\hat{c})^t$. Then, w.p. greater than $1 - \frac{2}{c^3 d^6}$, we have:*

$$y_i f(x_i) = \frac{t\eta}{4} \pm \frac{c_n}{\sqrt{\log d}} = (t \pm \text{\textonehalf})\frac{\eta}{4} \quad \forall (x_i, y_i) \in \mathcal{S}_n \tag{5}$$

*Proof.* We use equations (2), (3) & (4) to obtain simplify the dot product between $w_i^{(t)}$ & $x_j$ and the indicator $\mathbb{1}\left\{w_i^{(t)} \cdot x_j \geq 0\right\}$. First, we show that the dot product between $w_i^{(t)}$ and $x_j$ is in the band $\frac{t\eta v_i y_j}{2} \pm \frac{c_n}{\sqrt{k \log d}}$ with high probability:

$$w_i^{(t)} \cdot x_j = w_{1i}^{(t)}y_j + w_{2i}^{(t)}\frac{y_j + 1}{2}\varepsilon_j + \bar{w}_i^{(t)} \cdot \bar{x}_j \overset{(a)}{=} w_{1i}^{(t)}y_j + w_{2i}^{(t)}\frac{y_j + 1}{2}\varepsilon_j + ||\bar{w}_i^{(t)}||Z_j$$

$$\overset{(b)}{=} w_{1i}^{(t)}y_j + w_{2i}^{(t)}\frac{y_j + 1}{2}\varepsilon_j \pm ||\bar{w}_i^{(t)}||\sqrt{8\alpha \log d} \qquad\qquad \text{w.p. } 1 - \frac{2}{d^6}$$

$$= \frac{t\eta v_i y_j}{2} \pm \frac{c_0(1+\hat{c})^t}{\sqrt{dk}\log d}(y_j + \frac{y_j+1}{2}\varepsilon_j) \pm \frac{c_0(1+\hat{c})^t}{\sqrt{k}\log d}\sqrt{8\alpha \log d} \qquad \text{via eq. 2, 3, 4}$$

$$\overset{(c)}{=} \frac{t\eta v_i y_j}{2} \pm \frac{2c_0(1+\hat{c})^t}{\sqrt{dk}\log d} \pm \frac{3\sqrt{\alpha}c_0(1+\hat{c})^t}{\sqrt{k}\log d} = \frac{t\eta v_i y_j}{2} \pm \frac{c_n}{\sqrt{k}\log d} \tag{6}$$

$$= (ty_j \pm \text{\textonequarter})\frac{\eta v_i}{2} \quad \text{when } d \geq \exp((\frac{8c_n}{\eta})^2) \tag{7}$$

where $(a)$ is because $\bar{w}_i^{(t)} \cdot \bar{x}_j = ||\bar{w}_i^{(t)}||\mathcal{N}(0,1)$, $(b)$ is via lemma 7 & $c > 1$, and $(c)$ is because $(y_j + \frac{y_j+1}{2}\varepsilon_j) < 2$. Next, when $d \geq \exp((\frac{8c_n}{\eta})^2)$, we can simplify $\mathbb{1}\left\{w_i^{(t)} \cdot x_j \geq 0\right\}$ as follows:

$$\mathbb{1}\left\{w_i^{(t)} \cdot x_j \geq 0\right\} \overset{eq.7}{\leq} \mathbb{1}\left\{(ty_j \pm \text{\textonequarter})\frac{\eta v_i}{2} \geq 0\right\} \leq \begin{cases} 1, & \text{if } t = 0 \\ 1, & \text{if } t > 0 \text{ and } y_j v_i \geq 0 \leq \mathbb{1}\{t = 0 \lor yv_i \geq 0\} \\ 0, & \text{if } t > 0 \text{ and } y_j v_i < 0 \end{cases} \tag{8}$$

We can now use equations (6) & (8) to show that $y_j f^{(t)}(x_j)$ is in the band $t\eta/4 \pm O(1/\sqrt{\log d})$ with high probability:

$$y_j f^{(t)}(x_j) = \sum_{i=1}^k y_j v_i \cdot \text{ReLU}(w_i \cdot x_j) = \sum_{i=1}^k v_i \mathbb{1}\{t = 0 \lor yv_i \geq 0\}\left(\frac{t\eta v_i}{2} \pm \frac{c_n}{\sqrt{k}\log d}\right)$$

$$= \sum_{i=1}^k \mathbb{1}\{t = 0 \lor yv_i \geq 0\}\left(\frac{t\eta}{2k} \pm \frac{c_n}{k\sqrt{\log d}}\right) \overset{(a)}{=} \begin{cases} \pm k\frac{c_n}{k\sqrt{\log d}}, & \text{if } t = 0 \\ \frac{k}{2}\left(\frac{t\eta}{2k} \pm \frac{c_n}{k\sqrt{\log d}}\right), & \text{if } t > 0 \end{cases} \tag{9}$$

$$= \frac{t\eta}{4} \pm \frac{c_n}{\sqrt{\log d}} \overset{(b)}{=} (t \pm \text{\textonehalf})\frac{\eta}{4}$$

where $(a)$ is due to $|\{v_i \mid v_i > 0\}| = |\{v_i \mid v_i < 0\}| = k/2$ and $(b)$ follows from $c_n/\sqrt{\log d} \leq \eta/8$ when $d \geq \exp((8c_n/\eta)^2)$ $\blacksquare$

**Lemma 3.** *If equations (2), (3) & (4) hold at timestep $t$, $d > \exp((\frac{4c_0 e^1}{\eta})^2)$ and $d/\log d > \sqrt{k}$, we have:*

$$\max_{|\delta| \leq |w_{2i}^{(t)}|} \frac{1}{||\bar{w}^t||}\varphi\left(\frac{w_{1i}^{(t)} + \delta}{||\bar{w}_i^{(t)}||}\right) \leq 1 \tag{10}$$

$$\left|\phi\left(\frac{w_{1i}^{(t)} + w_{2i}^{(t)}}{||\bar{w}^{(t)}||}\right) - \phi\left(\frac{w_{1i}^{(t)} - w_{2i}^{(t)}}{||\bar{w}^{(t)}||}\right)\right| \leq \frac{2c_0(1+\hat{c})^t}{\sqrt{dk}\log d} \tag{11}$$

$$\left|\phi\left(\frac{w_{1i}^{(t)} + w_{2i}^{(t)}}{||\bar{w}^{(t)}||}\right) - \phi\left(\frac{w_{1i}^{(t)}}{||\bar{w}^{(t)}||}\right)\right| \leq \frac{c_0(1+\hat{c})^t}{\sqrt{dk}\log d} \tag{12}$$

$$\frac{1}{||\bar{w}^t||}\varphi\left(\frac{w_{1i}^{(t)}}{||\bar{w}_i^{(t)}||}\right) \leq \frac{c_0(1+\hat{c})^t}{\sqrt{dk}\log d} \tag{13}$$

*Proof.* Let $g_z(x) = \frac{1}{x}\varphi(\frac{z}{x})$ and $h(x) = \max_{|\delta| \le |w_{2i}^{(t)}|} \frac{1}{x}\varphi(\frac{w_{1i}^{(t)}+\delta}{x}) = \max_{|\delta| \le |w_{2i}^{(t)}|} g_{w_{1i}^t+\delta}(x)$. To prove Equation (10), we show that an upper bound on $||w^{(t)}||$ is less than a lower bound on $\arg\max_x h(x)$, which subsequently implies that $h(||w^{(t)}||) < \max_x h(x)$ because $h$ is an increasing function for all $|x| \le \arg\max_x h(x)$.

First, we find the maximizer $x^*$ of $h(x)$ as follows:

$$\max_x h(x) = \max_x \max_{|\delta| \le |w_{2i}^{(t)}|} g_{w_{1i}^{(t)}+\delta}(x) \overset{(a)}{=} \max_{|\delta| \le |w_{2i}^{(t)}|} \frac{e^{-1}}{|w_{1i}^{(t)}+\delta|} \qquad \text{when } x^* = |w_{1i}^{(t)}+\delta|$$

where $(a)$ follows from lemma 11. Next, we lower bound the maximizer $x^*$ of $h(x)$:

$$x^* = |w_{1i}^{(t)}+\delta| \ge |w_{1i}^{(t)}+w_{2i}^{(t)}| \ge \left||w_{1i}^{(t)}| - |w_{2i}^{(t)}|\right| \overset{(a)}{\ge} \left||\frac{t\eta v_i}{2} \pm \frac{c_0(1+\hat{c})^t}{\sqrt{dk}\log d}| - |\frac{c_0(1+\hat{c})^t}{\sqrt{dk}\log d}|\right|$$

$$\overset{(b)}{\ge} \left||\frac{t\eta v_i}{2} \pm \frac{\eta v_i}{8}| - |\frac{\eta v_i}{8}|\right| \ge \left|t - 1/2\right|\frac{\eta v_i}{2} \ge \frac{\eta v_i}{4}$$

where $(a)$ follows from the weights in the linear and slab coordinate at timestep $t$ (equations (2) & (3)) and $(b)$ is because $\frac{c_0(1+\hat{c})^t}{\sqrt{dk}\log d} \le \frac{\eta v_i}{8}$ when $\sqrt{d} \ge \frac{8c_0e^1}{\eta}$. Therefore, $\arg\max_x h(x) \ge \eta v_i/4$. We can use the upper bound on the $\ell_2$ norm of the gradient along the noise coordinates (equation (4)) and $d \ge \exp(\frac{4c_0e^1}{\eta})$ to show that $||\bar{w}_i^{(t)}||$ is less than $x^*$:

$$||\bar{w}_i^{(t)}|| \le \frac{c_0(1+\hat{c})^t}{\sqrt{dk}\log d} \le \frac{\eta v_i}{4} \le \arg\max_x h(x)$$

From lemma 11, we know that $h(x)$ is an increasing function for all $|x| < x^*$. This implies that $h(||w_i^{(t)}||) \le h(\frac{c_0(1+\hat{c})^t}{\sqrt{dk}\log d}) \le h(\frac{\eta v_i}{4}) \le h(x^*)$. Therefore, when $d \ge \exp((\frac{4c_0e^1}{\eta})^2)$ and $d/\log d \ge \sqrt{k}$, we obtain the desired result as follows:

$$\max_{|\delta| \le |w_{2i}^{(t)}|} \frac{1}{||\bar{w}^t||}\varphi\Big(\frac{w_{1i}^{(t)}+\delta}{||\bar{w}_i^{(t)}||}\Big) = h(||w_i^{(t)}||) \le h(\frac{c_0e^1}{\sqrt{dk}\log d}) \le \frac{\sqrt{k}\log d}{c_0e^1}\frac{1}{d^{(\frac{\eta}{4c_0e^1})^2\log d}} \le 1$$

Now, we can prove equations (11), (12) and (13) using equation (10) as follows:

$$\left|\phi\Big(\frac{w_{1i}^{(t)}+w_{2i}^{(t)}}{||\bar{w}^{(t)}||}\Big) - \phi\Big(\frac{w_{1i}^{(t)}-w_{2i}^{(t)}}{||\bar{w}^{(t)}||}\Big)\right| \le 2|w_{2i}^{(t)}| \cdot \max_{|\delta| \le |w_{2i}^{(t)}|} \frac{1}{||\bar{w}^t||}\varphi\Big(\frac{w_{1i}^{(t)}+\delta}{||\bar{w}_i^{(t)}||}\Big) \overset{(a)}{\le} \frac{2c_0(1+\hat{c})^t}{\sqrt{dk}\log d} \qquad (14)$$

$$\left|\phi\Big(\frac{w_{1i}^{(t)}+w_{2i}^{(t)}}{||\bar{w}^{(t)}||}\Big) - \phi\Big(\frac{w_{1i}^{(t)}}{||\bar{w}^{(t)}||}\Big)\right| \le |w_{2i}^{(t)}| \cdot \max_{|\delta| \le |w_{2i}^{(t)}|} \frac{1}{||\bar{w}^t||}\varphi\Big(\frac{w_{1i}^{(t)}+\delta}{||\bar{w}_i^{(t)}||}\Big) \overset{(a)}{\le} \frac{c_0(1+\hat{c})^t}{\sqrt{dk}\log d} \qquad (15)$$

$$\frac{1}{||\bar{w}^t||}\varphi\Big(\frac{w_{1i}^{(t)}}{||\bar{w}_i^{(t)}||}\Big) \le \max_{|\delta| \le |w_{2i}^{(t)}|} \frac{1}{||\bar{w}^t||}\varphi\Big(\frac{w_{1i}^{(t)}+\delta}{||\bar{w}_i^{(t)}||}\Big) \overset{(a)}{\le} \frac{c_0(1+\hat{c})^t}{\sqrt{dk}\log d} \qquad (16)$$

where $(a)$ is due to equation (4). ∎

### F.2 Closed-form Gradient Expressions

In this section, we provide closed-form expressions for gradients along the linear, slab and noise coordinates: $\nabla_{w_{1i}}\mathcal{L}_f(\mathcal{S}_n)$, $\nabla_{w_{2i}}\mathcal{L}_f(\mathcal{S}_n)$ and $\nabla_{\bar{w}_i}\mathcal{L}_f(\mathcal{S}_n)$. First, we provide a closed-form expression for the gradient along the linear coordinate:

**Lemma 4.** *If $n > cd^2$ and $y_i f(x_i) < 1 \, \forall (x_i, y_i) \in \mathcal{S}_n$, then w.p. greater than $1 - \frac{3}{n}$:*

$$\nabla_{w_{1i}}\mathcal{L}_f(\mathcal{S}_n) = -\frac{v_i}{4}\left[2 + \phi\Big(\frac{w_{1i}+w_{2i}}{||\bar{w}_i||}\Big) + \phi\Big(\frac{w_{1i}-w_{2i}}{||\bar{w}_i||}\Big) - 2\phi\Big(\frac{w_{1i}}{||\bar{w}_i||}\Big)\right] \pm \frac{5v_i}{d}\sqrt{\frac{\log(cd^2)}{c}}$$

*Proof.*

$$\nabla_{w_{1i}}\mathcal{L}_f(\mathcal{S}_n) = -\frac{v_i}{n}\sum_{j=1}^{n}\mathbb{1}\{y_j f(x_j) \le 1\}\mathbb{1}\left\{w_i^T x_j \ge 0\right\}y_j x_{1j}$$

$$\overset{(a)}{=} -\frac{v_i}{n}\sum_{j=1}^{n}\mathbb{1}\left\{\bar{w}_i^T \bar{x}_j \ge -w_{i1}y_j - w_{i2}\mathbb{1}\{y_j=1\}\varepsilon_j\right\}$$

$$\overset{(b)}{=} -\frac{v_i}{n}\sum_{j=1}^{n}\mathbb{1}\left\{Z_j \ge \frac{-w_{1i}y - w_{2i}\mathbb{1}\{y_j=1\}\varepsilon_j}{||\bar{w}_i||}\right\} \qquad\qquad \text{where } Z_j \sim \mathcal{N}(0,1)$$

$$= -v_i \sum_{l}^{\{0,1,-1\}} \frac{1}{n}\sum_{i=1}^{n}\mathbb{1}\{x_{2j}=l\}\mathbb{1}\left\{Z_j \ge \frac{-w_{1i}(2l^2-1)-w_{2i}l}{||\bar{w}_i||}\right\}$$

$$= -v_i \sum_{l}^{\{0,1,-1\}} \left(\mathbb{P}(x_{2j}=l)\phi\Big(\frac{w_{1i}(2l^2-1)+w_{2i}l}{||\bar{w}_i||}\Big) \pm \sqrt{\frac{\log n}{n}}\right) \qquad\qquad \text{via lemma 9}$$

$$= -\frac{v_i}{4}[\phi(\frac{w_{1i}+w_{2i}}{||\bar{w}_i||}) + \phi(\frac{w_{1i}-w_{2i}}{||\bar{w}_i||}) + 2\phi(\frac{-w_{1i}}{||\bar{w}_i||})] \pm \frac{5v_i}{d}\sqrt{\frac{\log(cd^2)}{c}} \qquad\qquad n > cd^2$$

$$= -\frac{v_i}{4}\left[2 + \phi\Big(\frac{w_{1i}+w_{2i}}{||\bar{w}_i||}\Big) + \phi\Big(\frac{w_{1i}-w_{2i}}{||\bar{w}_i||}\Big) - 2\phi\Big(\frac{w_{1i}}{||\bar{w}_i||}\Big)\right] \pm \frac{5v_i}{d}\sqrt{\frac{\log(cd^2)}{c}} \qquad\qquad \text{w.p. } 1 - \frac{3}{n}$$

where $(a)$ is due to $y_i x_{i1} = y_i^2 = 1$ & $\mathbb{1}\{y_j f(x_j) \le 1\} = 1$ and $(b)$ is due to $\mathbb{1}\{\bar{w}_i^T \bar{x}_j \ge k\} = \mathbb{1}\{||\bar{w}_i||Z_j \ge k\}$. ∎

Similarly, we provide a closed-form expression for the gradient along the slab coordinate:

**Lemma 5.** *If $n > cd^2$ and $y_i f(x_i) < 1 \forall (x_i, y_i) \in \mathcal{S}_n$, then w.p. greater than $1 - \frac{3}{n}$:*

$$\nabla_{w_{2i}}\mathcal{L}_f(\mathcal{S}_n) = -\frac{v_i}{4}\left[\phi\Big(\frac{w_{1i}+w_{2i}}{||\bar{w}_i||}\Big) - \phi\Big(\frac{w_{1i}-w_{2i}}{||\bar{w}_i||}\Big)\right] \pm \frac{5v_i}{d}\sqrt{\frac{\log(cd^2)}{c}}$$

*Proof.*

$$\nabla_{w_{2i}}\mathcal{L}_f(\mathcal{S}_n) = -\frac{v_i}{n}\sum_{j=1}^{n}\mathbb{1}\{y_j f(x_j) \le 1\}\mathbb{1}\left\{w_i^T x_j \ge 0\right\}y_j x_{2j}$$

$$\overset{(a)}{=} -\frac{v_i}{n}\sum_{j=1}^{n}\mathbb{1}\left\{\bar{w}_i^T \bar{x}_j \ge -w_{1i}y_j - w_{2i}\mathbb{1}\{y_j=1\}\varepsilon_j\right\}\mathbb{1}\{y_j=1\}\varepsilon_j$$

$$\overset{(b)}{=} v_i \sum_{l}^{\{-1,1\}} \frac{1}{n}\sum_{j=1}^{n}(-1)^{\mathbb{1}\{\varepsilon_j=l\}}\mathbb{1}\left\{Z_j \ge \frac{-w_{1i}-w_{2i}l}{||\bar{w}_i||}\right\}$$

$$= -v_i \sum_{l}^{\{1,-1\}} \left(\mathbb{P}(x_{2j}=l)\phi\Big(\frac{w_{1i}+w_{2i}l}{||\bar{w}_i||}\Big) \pm \sqrt{\frac{\log n}{n}}\right) \qquad\qquad \text{via lemma 9}$$

$$= -\frac{v_i}{4}[\phi(\frac{w_{1i}+w_{2i}}{||\bar{w}_i||}) - \phi(\frac{w_{1i}-w_{2i}}{||\bar{w}_i||})] \pm \frac{5v_i}{d}\sqrt{\frac{\log(cd^2)}{c}} \qquad\qquad n > cd^2$$

$$= -\frac{v_i}{4}\left[\phi\Big(\frac{w_{1i}+w_{2i}}{||\bar{w}_i||}\Big) - \phi\Big(\frac{w_{1i}-w_{2i}}{||\bar{w}_i||}\Big)\right] \pm \frac{5v_i}{d}\sqrt{\frac{\log(cd^2)}{c}} \qquad\qquad \text{w.p. } 1 - \frac{3}{n}$$

where $(a)$ is due to $y_i x_{i2} = \mathbb{1}\{y_i=1\}\varepsilon_i$ & $\mathbb{1}\{y_j f(x_j) \le 1\} = 1$ and $(b)$ is due to $\mathbb{1}\{\bar{w}_i^T \bar{x}_j \ge k\} = \mathbb{1}\{||\bar{w}_i||Z_j \ge k\}$. ∎

Next, we provide a closed-form expression for the gradient along the noise coordinates:

**Lemma 6.** *If $n > cd^2$ and $y_i f(x_i) < 1 \,\forall (x_i, y_i) \in \mathcal{S}_n$, then w.p. greater than $1 - \frac{1}{3n}$:*

$$\nabla_{\bar{w}_i} \mathcal{L}_f(\mathcal{S}_n) = \bar{\mathcal{G}} \bar{w}_i \pm \frac{3|v_i| \log(\sqrt{cd})}{\sqrt{cd}} \frac{\bar{w}_i}{||\bar{w}_i||} \pm \frac{6|v_i|}{\sqrt{cd}} u_i^\perp$$

$$\bar{\mathcal{G}} = -\frac{v_i}{4||\bar{w}_i||} \left[ \varphi\left(\frac{w_{1i} + w_{2i}}{||\bar{w}_i||}\right) + \varphi\left(\frac{w_{1i} - w_{2i}}{||\bar{w}_i||}\right) - 2\varphi\left(\frac{w_{1i}}{||\bar{w}_i||}\right) \right]$$

*where $u_i^\perp$ is some unit vector orthogonal to $\bar{w}_i$.*

*Proof.* Let $S \subset \mathbb{R}^{d-2}$ denote the subspace spanned by $\bar{w}_i$. Then, for any $x \in \mathbb{R}^d$, $x = x^S + x^{S^\perp}$ where $x^S$ & $x^{S^\perp}$ are the orthogonal projections of $x$ onto $S$ and its orthogonal complement $S^\perp$. We show the $\ell_2$ norm of the orthogonal projections of $\nabla_{\bar{w}_i} \mathcal{L}_f(\mathcal{S}_n)$ onto $S$ and $S^\perp$ are $O(\frac{1}{\sqrt{d}})$:

$$\nabla_{\bar{w}_i} \mathcal{L}_f(\mathcal{S}_n) = -\frac{v_i}{n} \sum_{j=1}^n \mathbb{1}\{y_j f(x_j) \leq 1\} \mathbb{1}\left\{w_i^T x_j \geq 0\right\} y_j \bar{x}_j$$

$$= \underbrace{-\frac{v_i}{n} \sum_{j=1}^n \mathbb{1}\left\{w_i^T x_j \geq 0\right\} y_j (\bar{x}_j^S)}_{\text{case 1}} \underbrace{-\frac{v_i}{n} \sum_{j=1}^n \mathbb{1}\left\{w_i^T x_j \geq 0\right\} y_j (\bar{x}_j^{S^\perp})}_{\text{case 2}}$$

Next, we show that the projection of $\nabla_{\bar{w}_i} \mathcal{L}_f(\mathcal{S}_n)$ onto $S^\perp$ (i.e., case 2) has small norm w.p. greater than $1 - \frac{1}{d}$:

$$||\nabla_{\bar{w}_i} \mathcal{L}_f(\mathcal{S}_n))^{S^\perp}|| = ||\frac{v_i}{n} \sum_{j=1}^n \mathbb{1}\left\{w_i^T x_j \geq 0\right\} y_j \bar{x}_j^{S^\perp}|| = ||\frac{v_i}{n} \sum_{j=1}^n \mathbb{1}\left\{w_i^T x_j^S \geq 0\right\} y_j \bar{x}_j^{S^\perp}||$$

$$\overset{(a)}{\leq} |v_i| \cdot ||\sum_{j=1}^n \mathcal{N}(0, \frac{1}{n^2} I_{d-2})|| = |v_i| \cdot ||\mathcal{N}(0, \frac{1}{n} I_{d-2})||$$

$$\overset{(b)}{\leq} 4|v_i| \sqrt{\frac{d}{n}} \pm 2|v_i| \sqrt{\frac{\log n}{n}} \overset{(c)}{\leq} \frac{6|v_i|}{\sqrt{cd}} \qquad\qquad \text{w.p. } 1 - \frac{1}{n}$$

where $(a)$ is because $x_j^S \perp \bar{x}^{S_j^\perp}$, $(b)$ is via fact 1 and $(c)$ is due to $n \geq cd^2$. Next, we show that the norm of the gradient in the direction of $\bar{w}_i$ (i.e., case 1) is close to $\bar{\mathcal{G}}$ w.h.p.:

$$\nabla_{\bar{w}_i} \mathcal{L}_f(\mathcal{S}_n))^S = -\frac{v_i}{n} \sum_{j=1}^n \mathbb{1}\left\{w_i^T x_j \geq 0\right\} y_j \bar{x}_j^S$$

$$\overset{(a)}{=} -\left(\frac{1}{n} \sum_{j=1}^n \mathbb{1}\left\{w_i^T x_j^S \geq 0\right\} y_j \bar{w}_i^T \bar{x}_j\right) \frac{v_i \bar{w}_i}{||\bar{w}_i||^2}$$

$$\overset{(b)}{=} -\left(\frac{1}{n} \sum_{j=1}^n \mathbb{1}\left\{Z_j \geq \frac{-w_{1i} y - w_{2i} \mathbb{1}\{y_j = 1\} \varepsilon_j}{||\bar{w}_i||}\right\} y_j Z_j\right) \frac{v_i \bar{w}_i}{||\bar{w}_i||}$$

$$= \left(\sum_{l}^{\{0, \pm 1\}} \frac{(-1)^{\mathbb{1}\{l \neq 0\}}}{n} \sum_{i=1}^n \mathbb{1}\left\{x_{2j} = l \wedge Z_j \geq \frac{-w_{1i}(2l^2-1) - w_{2i} l}{||\bar{w}_i||}\right\} Z_j\right) \frac{v_i \bar{w}_i}{||\bar{w}_i||}$$

$$= \left[2\varphi(\frac{w_{i1}}{||\bar{w}_i||}) - \varphi(\frac{w_{1i} + w_{2i}}{||\bar{w}_i||}) - \varphi(\frac{w_{1i} - w_{2i}}{||\bar{w}_i||}) \pm \frac{5 \log n}{\sqrt{n}}\right] \frac{v_i \bar{w}_i}{4||\bar{w}_i||} \qquad \text{via lemma 10}$$

$$= \bar{\mathcal{G}} \bar{w}_i \pm \frac{3|v_i| \log(\sqrt{cd})}{\sqrt{cd}} \frac{\bar{w}_i}{||\bar{w}_i||} \qquad\qquad \text{w.p. } 1 - \frac{12}{n}$$

where $(a)$ is because $\bar{x}_j^S = \frac{\bar{w}_i^T x_j}{||\bar{w}_i||^2} \bar{w}_i$ and $(b)$ is because $(b)$ is due to $\mathbb{1}\{\bar{w}_i^T \bar{x}_j \geq k\} = \mathbb{1}\{||\bar{w}_i|| Z_j \geq k\}$. Therefore, by combining the results in case 1 and 2, the following holds w.p. greater than $1 - \frac{13}{n}$:

$$\nabla_{\bar{w}_i f}(\mathcal{S}_n) = \bar{\mathcal{G}} \bar{w}_i \pm \frac{3|v_i| \log(\sqrt{cd})}{\sqrt{cd}} \frac{\bar{w}_i}{||\bar{w}_i||} \pm \frac{6|v_i|}{\sqrt{cd}} u_i^\perp$$

$\blacksquare$

## F.3 Miscellaneous Lemmas

**Lemma 7.** *Let $X_i \sim \mathcal{N}(0, \sigma^2)$ and $\delta \in (0, 1)$. Then, $\max_{i \in [k]} |X_i| \leq \sigma \sqrt{2 \log(\frac{2k}{\delta})}$ with probability greater than $1 - \delta$,*

*Proof.* Let $\varphi$ denote the probability density function of the standard normal. Also let $Z \sim \mathcal{N}(0, 1)$. Then, for $t \geq 1$, we have:

$$\mathbb{P}(|X| \geq \sigma t) = \mathbb{P}(|Z| \geq t) = 2 \int_t^\infty x \varphi(x) \, dx \leq \frac{2}{t} \int_t^\infty x \varphi(x) \, dx \overset{(a)}{\leq} \frac{2}{t} \int_\infty^t \varphi'(x) \, dx \leq \frac{2}{t} \varphi(t) \leq 2\varphi(t)$$

where $(a)$ is because $\varphi'(x) = -x\varphi(x)$. Using union bound with $t = \sqrt{2 \log(\frac{2k}{\delta})} \geq 1 \, \forall \delta \in (0, 1)$ gives the desired result. ∎

**Lemma 8.** *Let $\phi$ and $\varphi$ denote the cumulative distribution function and the probability density function of the standard gaussian. Then, for any $Z \sim \mathcal{N}(0, 1)$ and $k \in \mathbb{R}$:*

$$\mathbb{E}[\mathbb{1}\{Z \geq k\} Z] = \varphi(k) = \exp(-k^2/2)$$

*Proof.* The expectation $\mathbb{E}[\mathbb{1}\{Z \geq k\} Z]$ can be simplified as follows:

$$\mathbb{E}[\mathbb{1}\{Z \geq c\} Z] = \Pr[Z \geq c] \mathbb{E}[Z | Z \geq c] = \phi^c(k) \int_k^\infty x \frac{\varphi(x)}{\phi^c(k)} \, dx \overset{(a)}{=} - \int_k^\infty \varphi'(x) \, dx = \varphi(k)$$

where $(a)$ is due to $\varphi'(x) = -x\varphi(x)$. ∎

**Lemma 9.** *Let $b_i \sim bernoulli(p)$ and $Z_i \sim \mathcal{N}(0, 1)$. Let $X_i = b_i \mathbb{1}\{Z_i \geq k\}$ and $\bar{X} = \frac{1}{n} \sum_{i=1}^n X_i$. Then:*

$$\Pr\left(|\bar{X} - p\phi(-k)| \geq \sqrt{\frac{\log n}{n}}\right) \leq \frac{1}{n}$$

*Proof.* Note that $\mathbb{E}[\bar{X}] = \mathbb{E}[X_i] = \mathbb{E}[b_i] \mathbb{E}[\mathbb{1}\{Z_i \geq k\}] = p\phi(-k)$ and $|X_i| \leq 1$. Therefore, using Hoeffding's inequality with $t = \sqrt{\frac{\log n}{n}}$ directly gives the result. ∎

**Lemma 10.** *Let $b_i \sim bern(p)$ and $Z_i \sim \mathcal{N}(0, 1)$. Let $X_i = b_i \mathbb{1}\{Z_i \geq k\} Z_i$ and $\bar{X} = \frac{1}{n} \sum_{i=1}^n X_i$. Then:*

$$\mathbb{P}\left(|\bar{X} - p\varphi(k)| \leq \sqrt{\frac{2}{n}} \log n\right) \geq 1 - \frac{4}{n}$$

*Proof.* Since $|X_i| = |b_i \mathbb{1}\{Z_i \geq k\} Z_i| \leq |Z_i|$, we have $\max_{i \in [n]} |X_i| \leq \sqrt{4 \log(n)}$ w.p. at least $1 - \frac{2}{n}$ via lemma 7. From lemma 8, we get $\mathbb{E}[X_i] = \mathbb{E}[b_i] \mathbb{E}[\mathbb{1}\{Z_i \geq k\} Z_i] = p\varphi(k)$. Let $A = \mathbb{1}\{|X_i| \leq \sqrt{4 \log(n)} \, \forall i \in [n]\}$. Given $A$, we can use Hoeffding's inequality with $t^* = \sqrt{\frac{2}{n}} \log n$ (and $\delta = 2/n$) to get the desired result, as follows:

$$\mathbb{P}(|\bar{X} - p\varphi(k)| \leq t^*) \geq \mathbb{P}(|\bar{X} - p\varphi(k)| \leq t^* | A) \mathbb{P}(A) \geq (1 - \frac{2}{n})^2 \geq 1 - \frac{4}{n}$$

Therefore, $\bar{X} = p\varphi(k) \pm \sqrt{\frac{2}{n}} \log n$ w.p. at least $1 - \frac{4}{n}$. ∎

**Lemma 11.** *Let $g : \mathbb{R} \backslash \{0\} \to \mathbb{R}$ be defined as $g_z(x) = \frac{1}{x} \exp(-\frac{z^2}{2x^2})$. Then, (1) $|z|$ and $-|z|$ are the global maximizer and minimizer respectively, and (2) $g$ monotonically increases from $-|z|$ to $|z|$.*

*Proof.* Note that $g_z'(x) = \frac{1}{x^2} \exp(-\frac{z^2}{2x^2})(\frac{z^2}{x^2} - 1)$. Therefore, the critical points of $g$ are $|z|$ and $-|z|$. Let $S = \{t : |t| \geq |z|, t \in \mathbb{R}/\{0\}\}$. Note that $g_z'(x) < 0$ for all $x \in S$ and $g_z'(x) > 0$ for all $x \in S^c$. Therefore, (1) and (2) hold. ∎

**Fact 1.** *Let $X \sim \mathcal{N}(0, \sigma^2 I_d)$ denote a d-dimensional gaussian vector. Then, from [75], w.p. greater than $1 - \delta$:*

$$||X||_2 \leq 4\sigma\sqrt{d} + 2\sigma$$