[Reviews · NeurIPS 2020]

Review 1

Summary and Contributions: The paper studies the implicit bias of neural networks, and argues that networks are biased to learn "simpler" features, even when they could be more accurate and robust by learning more complex ones. The main contribution of this paper is introducing several simplified data distributions where this can be more easily investigated. They show that certain neural networks empirically learn only "simple" features on these distributions, and also show this theoretically in a toy setting.

Strengths: The main strength of the work is in the data-distributions which they introduce. These are a simple and interesting testbed for theories about inductive bias, and they could be used in future work in the area. The "CIFAR+MNIST" dataset is especially interesting, since it is a more realistic distribution, and clearly demonstrates the "simplicity bias." The theoretical section is also a nice observation, extending existing results to the newly-introduced distributions. This paper will be relevant to the NeurIPS community, since it sheds more light on the implicit bias of neural networks (empirically and theoretically).

Weaknesses: The main weakness is that most of the settings studied are toy models: synthetic non-image distributions (with the exception of the CIFAR/MNIST experiment). The effects in these toy models are interesting, but it is unclear how much they say about implicit bias on real distributions.

Correctness: The theoretical claims appear to be correct. The proof of Theorem 1 could use some elaboration: specifically, the conclusion about small test error is not clearly demonstrated (the relevant Lemma F.3 appears to be about the train loss, without connecting it to the test loss). I don't think this will affect correctness of the Theorem, although the proof should be clarified.

Clarity: The paper is overall written reasonably, though there are some points where claims are overstated. Most notably, the term "explains" is overused (eg: SB "explains" adversarial examples, distribution shift, spurious features, etc). The examples in this paper do take steps towards understanding these phenomena, but it is too strong to claim they "explain" them. After all, this paper considers toy distributions -- and further, several of the terms above are not formally defined in the literature (eg "spurious features"), so it is unclear what an explanation would entail.

Relation to Prior Work: Prior work is sufficiently discussed in context. This problem of implicit bias is studied in various ways, and the field is young enough that every paper typically studies it in a different way.

Reproducibility: Yes

Additional Feedback: Comments which do not affect the review score: I suggest cutting down on the number of "synthetic distributions" introduced in the main body. Currently there are 4 synthetic distributions, but all share roughly the same interesting feature, and it's not worth forcing the reader to context-switch between these in order to make points (which could, presumably, be made equally well just by focusing on 1 distribution). The CIFAR/MNIST distribution is in my opinion the most interesting aspect of this work, and could be highlighted more. Section 5, that the implicit bias can hurt generalization, is interesting -- one would imagine that fully-connected nets can exploit the non-linear component. This could be worthy of more discussion. Section 4.3 is a fairly weak section, in that most of it is speculation that is not truly supported by the data. I suggest moving it out of the main body, or putting it into a "discussion" section explicitly. There is also a lot of interesting-looking material in the appendix. Consider adding a sketch of appendix-results towards the end of the main body. ======= Post-rebuttal update: I reduced my score by 1 pt because the concerns about over-claiming were not adequately addressed. However, I would like to see this paper appear in NeurIPS if these concerns are addressed.


Review 2

Summary and Contributions: [UPDATE AFTER REBUTTAL] I'm generally excited about this paper! The authors addressed my suggestions by stating that the code & dataset will be open-sourced, and by describing two additional experiments studying the role of initialization as well as a different loss function. Probably due to time constraints, the authors do not provide any results for these experiments in the rebuttal. I hope these experiments will be included in the final version. Given the author response I see no reason to down-grade my score; at the same time concerns raised by other reviewers regarding overly broad claims prevent me from raising my score (but I think these concerns can be quite easily addressed). I thus support publication at NeurIPS. [ORIGINAL REVIEW] The paper "The Pitfalls of Simplicity Bias in Neural Networks" investigates the tendency of neural networks to learn "simple" solutions. More specifically, it shows that CNNs a) ignore complex features when simple ones are equally predictive, b) ignore complex features even when simple ones are less predictive and c) base their "confidence" mostly on simple features. The paper proposes a number of simple toy datasets where these phenomena can be studied and convincingly shows the prevalence and scope of the "simplicity bias". The implications of simplicity bias are often considered to be beneficial in the sense that this prevents overfitting, but this paper shows how it can lead to poor OOD generalisation and how models often fall short from learning more than the most simple solution.

Strengths: The paper has a number of experiments with different architectures and optimizers (SGD, Adam), providing support to the empirical evaluation. The findings are surprising (not necessarily that these pitfalls exist but that they are so strong) and very relevant to the NeurIPS community. Overall, the observed problematic - in combination with other works in this direction - underlie one of the most important problems of present-day deep learning. A better understanding of these phenomena is a necessity, and the paper does a very good job in providing a better understanding. I believe it will be a valuable contribution to the community.

Weaknesses: - Simplicity bias is attributed to "standard training procedures such as SGD". However, Jacobsen et al. (reference [20]) showed that cross-entropy may be to blame: a modified loss function encourages neural networks to learn more than the most simple solution. This issue should be discussed and ideally one would like to investigate the invertible network from [20] on the proposed datasets. - The paper could benefit from investigating the role of initialization: clearly, a model that happened to be initialized such that a complex feature is already "learned" would make use of that feature. But at which point would it switch to learn the simpler one? I.e., one could design an experiment where a network is trained on a dataset where only the complex feature is predictive. This network is then used as the initialization for a dataset where both the complex and simpler feature are predictive. Linearly interpolating between the weights of this network and a random initialization could enable one to investigate how much the weights can deviate from the "complex feature solution" such that the complex solution is still learned, or whether the simple solution will always be preferred irrespective of initialization. - I appreciate that code was submitted alongside the paper. That being said, it would be good to mention if/how the dataset will be made available to others (which would be very helpful).

Correctness: I have not thoroughly checked the math, but the overall approach and datasets look convincing and make sense to me.

Clarity: The paper is very well written and figures nicely illustrate the setup. The appendix does not comply with the NeurIPS style file. At points the paper appears a bit crammed.

Relation to Prior Work: The paper discusses prior and related work. However, many aspects seem related to "shortcut learning" (https://arxiv.org/abs/2004.07780) and I believe the reader will benefit from discussing this connection. At some point, the authors mention that "to the best of our knowledge, prior works only focus on the positive aspect of SB: the lack of overfitting in practice". This is not the case e.g. in the shortcut paper, where the problematic aspects of SB are discussed as well. That being said, I agree with the authors that the positive aspects of SB have been more prevalent in the literature, so re-phrasing this statement accordingly would be more accurate.

Reproducibility: Yes

Additional Feedback: - line 274: point missing. - concurrent work (https://arxiv.org/pdf/2006.12433.pdf) observes that "when two features redundantly predict the label, the model preferentially represents one, and its preference reflects what was most linearly decodable from the untrained model." Just FYI in case the authors haven't seen this already, this may be a pattern worth looking out for in the future.


Review 3

Summary and Contributions: The paper develops a notion of feature simplicity. It designs datasets and shows NNs rely on simple features in these. This is offered as an explanation why NNs have poor robustness to data shift.

Strengths: The paper addresses the important questions of robustness in neural networks and generalization, specifically its relationship to simplicity. It introduces synthetic datasets which allow both theoretical and empirical evaluation of simplicity and test errors. The contributions are novel to my knowledge, and ambitious in their scope.

Weaknesses: Several key claims seem false or insufficiently supported. 1) The authors demonstrate that “contrary to conventional wisdom, extreme [simplicity bias] can in fact hurt generalization”. We can always _construct_ a dataset where a simplicity-biased algorithm will fail. This is unsurprising and follows from the no free lunch theorems. Constructing datasets is in fact the method used here. But the practical significance/realism of these datasets must be established. This is not done at all for the first 3 datasets. The MNIST-CIFAR dataset may be more realistic but on it’s own it is not enough to establish the paper’s claims. 2) The authors show that simplicity bias is one possible reason for non-robustness. But they claim that simplicity bias is the cause behind non-robustness in other datasets. This is a large leap of faith with no support. 3) Key claims are phrased as if they apply to neural networks in general. In fact, theoretical results seem to be limited to the synthetic constructed datasets, and do not seem to apply generally. Nonetheless, this work could be promising if the relevance and generality of the bias towards "simple" features was established. The observation that NNs can learn small-margin classifiers if they correspond to simple features is interesting, though it is unclear if it holds outside the constructed datasets.

Correctness: As noted above, the methods do not sufficiently support the key claims, but I have not spotted any technical errors.

Clarity: The abstract and introduction are clear and appear promising. Section 3.0 neglects to define a few terms (see below) and from section 4.3 the paper becomes less focused.

Relation to Prior Work: A discussion of simplicity is missing. I personally think this would be more useful than discussing distribution shift and adversarial robustness.

Reproducibility: Yes

Additional Feedback: - Is MNIST necessarily less complex than CIFAR-10? It’s intuitive but not otherwise justified. - Definition of simplicity: distinguish simplicity of functions, which is more standard, vs features - Claims are overly general, implying that they apply to all NNs (L74-87) - Related work on simplicity and SB is missing - Claim (ii) :SB is only one explanation for overconfidence, others seem possible - L135-146 are hard to follow. For example, it would be useful to define x^S, x^Sc, \bar{x}^S. The same goes for “marginal distribution of S”, I don’t think the distribution of a set is meant here? - The abstract states that a key shortcoming in prior work is that simplicity is vaguely defined. The authors claim that the number of linear classifiers needed is a “natural notion of simplicity” but this is not justified. Furthermore, no method is offered to measure it (except in the authors own datasets by construction). - Typo: “would almost perfect” Edit: I increased my score by one point since the authors now gave some argument why their datasets are realistic (although not a decisive one in my view). Additionally, the authors indicated they will acknowledge that their results apply to specific, constructed datasets and not necessarily in general.


Review 4

Summary and Contributions: ** Update after authors feedback ** I will maintain my score and recommend this work for acceptance. I do however agree with some of the other reviews that some parts of the paper could be written more cautiously and would like to encourage the authors to do so. --- In this work the authors describe an approach to systematically investigate a certain kind of bias and therefore also the generalization properties of neural networks for supervised classification. At its core, the authors propose to construct datasets that contain both “simple” and more “complex” features which are all highly predictive for the given task. In a sequence of experiments the authors show convincingly that fully connected MLPs, ConvNets and GRU based sequence models all almost exclusively concentrate on the simple features in the input data to make predictions, disregarding the more complex but equally predictive features in the data. The authors call this property the “simplicity bias”. Besides empirically demonstrating this property for a range of neural network architectures and for various optimization methods (SGD, Adam RMSProp), the authors furthermore present a proof that this is expected for one hidden layer neural networks with ReLU activations.

Strengths: The authors start with a simple idea, the definition of “simple” and “complex” predictive features, construct datasets from it and use these to systematically investigate the sensitivity of the learned neural networks to these input features. This simple approach proves to be strikingly powerful. The paper and the appendix describe a vast number of experiments that together form a very convincing argument that the trained networks indeed exhibit a very strong simplicity bias. The large number of systematically conducted experiments make very unlikely that the observed effects are the result of any one of the specific architectural or hyperparameter choices, but rather an inherent feature of the way we currently train neural networks. The paper furthermore describes an impressive proof that this is indeed expected in one hidden layer neural network. Besides providing immediate insight into the properties of neural networks as they are trained today, I suspect that this paper will spark significant follow up work which might further investigate, or try to mitigate the described and harmful simplicity bias. m

Weaknesses: I generally don’t see any weaknesses in the work as it is presented here. The authors make effective use of the 8 page limit and the space in the appendix. The presented methodology suggests and enables future work that might investigat other aspects related to the effects described here. For example, it would be interesting to see whether techniques like drop-out or batch-norm have any influence on the obtained results. But at this point, I think it is absolutely acceptable to leave such questions for follow-up work.

Correctness: The presented arguments appear sound, the experiments are systematic and thorough.

Clarity: The paper is well written, easy to follow and makes effective use of the space available. The appendix contains a large number of additional experiments, details and insights.

Relation to Prior Work: In section 2 the authors correctly point out that the work presented here touches upon previous work on generalization, out-of distribution performance and adversarial robustness. I think prior work in this area is adequately discussed.

Reproducibility: Yes

Additional Feedback:

[Author Response · NeurIPS 2020]

We thank all reviewers for their insightful comments. Reviewer-specific comments follow.

Reviewer 1

**"unclear how much...on real distributions"**: We will improve the writing in the revised version to explicitly note that
our results are in the context of the proposed datasets. More importantly, note that the key design characteristic of
our slab-structured datasets—*multiple independent features of varying predictive power and simplicity*—is motivated
by recent empirical findings that differentially characterize learned features and desired features: syntactic cues vs.
semantic meaning [27], non-robust vs. robust features [19] and statistical regularities vs. high-level concepts [21].
**Regarding Theorem 1**, Lemma F.3 actually provides a precise description about the classifier after the $t^{th}$ mini-batch
gradient descent step. We subsequently use this characterization to bound the train and test loss in Lemma F.2.
**Section 4.3 and Appendix**: Thank you for your suggestions—we shall shorten section 4.3 and add a brief description
of our results presented in the appendices in the revised version.

Reviewer 2

**Role of initialization**: Our results on simplicity bias are robust to the exact choice of *random* initialization. That said,
we agree that understanding the tradeoff between (non-random) initialization and simplicity bias is an important next
step—we plan to study this tradeoff by varying the extent to which the model (at initialization) is aligned with the
complex slab features in the slab-structured and MNIST-CIFAR datasets.
**Amended Cross Entropy**: Thank you for this suggestion. We are in the process of porting the open-sourced code
(https://github.com/jhjacobsen/fully-invertible-revnet) to train invertible neural networks with amended cross entropy
on the synthetic datasets and will add the results to Appendix E in the revised version.
**Related work and Code**: Thank you for sharing relevant works. We will discuss connections to both papers—shortcut
learning (arXiv:2004.07780) and concurrent work (arXiv:2006.12433)—in the revised version. We shall open-source
our code and datasets as well.

Reviewer 3

**Slab-structured datasets**: Please see the response to Reviewer 1. Our focus on distilling empirical findings on real
datasets into synthetic datasets clearly sets us apart from pathological worst-case data distributions used to prove
no-free-lunch theorems. We believe that the slab-structured datasets can be used (a) as testbeds to develop algorithms
that improve the robustness of neural networks and (b) to gain theoretical insights that cannot be obtained from "linearly
separable" data [4], as they do not capture the failure modes of NNs observed in practice. Several papers have studied
such "principled synthetic datasets" to obtain insights on initial learning rate [R1], adversarial examples [R2], long-term
dependencies [R3], GAN dynamics [R4], generalization vs. robustness [R5], and bias in generative models [R6].
**Regarding overly general claims**: We would like to clarify our viewpoint: we show that simplicity bias is *an*
important factor that *jointly*, but not exclusively, contributes to the adversarial vulnerability, poor OOD performance and
suboptimal generalization (see lines 13, 92, 261, 296). We agree with reviewer 3—the paper should not give readers an
impression that our claims hold for *all* datasets and for *every* setting. To address this misunderstanding, we will rectify
our writing in the revised version to explicitly note that our empirical & theoretical results hold in the context of the
proposed slab-structured and MNIST-CIFAR datasets.
**Defining simplicity**: Regarding "natural notion of simplicity", our definition (line 161) is equivalent to the minimal
width of one-hidden-layer ReLU NNs required to perfectly fit a given dataset. "Number of linear classifiers...not
justified": The number of pieces in piecewise linear functions determines the VC dimension of this concept class (e.g.,
see [R7]). "mnist necessarily less complex than cifar10?" [R8] use spectrally normalized margin distributions to show
that mnist is less complex than cifar10. Moreover, linear models trained on the mnist & cifar blocks individually attain
99.9% & 68.9% test accuracy respectively, even though the cifar block is almost fully predictive of its labels (line 251).

Reviewer 4

**"...if drop-out or batch-norm have any influence on the obtained results"**: In Appendix C, we validate our results
on extreme simplicity bias across architectures, activation functions, optimizers and regularization methods such as L2
regularization and dropout. We will add another section on the effect of batch-norm in the revised version.

[R1] Li, Yuanzhi, et al. "Towards explaining regularization effect of initial large learning rate in neural networks." NeurIPS (2019).

[R2] Gilmer, Justin, et al. "Adversarial spheres." arXiv:1801.02774 (2018).

[R3] Hochreiter, Sepp, and Jürgen Schmidhuber. "Long short-term memory." Neural computation (1997)

[R4] Li, Jerry, et al. "On the limitations of first-order approximation in gan dynamics." ICML (2018).

[R5] Raghunathan, Aditi, et al. "Understanding and mitigating the tradeoff between robustness and accuracy." ICML (2020).

[R6] Zhao, Shengjia, et al. "Bias and generalization in deep generative models: An empirical study." NeurIPS (2018).

[R7] Bartlett, P., et al. "Nearly-tight VC-dimension & pseudodimension bounds for piecewise linear neural networks." JMLR (2019).

[R8] Bartlett, P., et al.. "Spectrally-normalized margin bounds for neural networks." NeurIPS (2017).


[Meta-Review · NeurIPS 2020]

The work aims at studying the inductive bias implicitly implemented by DNNs trained with SGD. The authors do so by introducing several synthetic toy and image-based datasets, where the notion of "simple feature" is made precise. By training DNNs on these datasets and analyzing the resulting models, the authors establish several curious observations: (Observation 1) When there is *one* simple feature and *many* less simple features, while each feature is predictive enough to allow 100% test accuracy, DNN ends up using *only* the simple feature; (Observation 2) Moreover, if there is *one* simple feature and *many* less simple features, so that the simplest features only allows 90% test accuracy while each of the less simple features leads to 100% test accuracy, DNNs end up using only the simplest feature (and thus suffering 90% test accuracy). These conclusions seem to hold for a range of model architectures and training scenarios. Moreover, the authors provide a theoretical result (Theorem 1) proving the aforementioned simplicity bias for 1-hidden layer DNNs trained with hinge loss on one of their specific synthetic distributions. Overall, the reviewers agree that the focus/topic of this paper are timely. The implicit bias of DNNs is one of the crucial ingredients of the "Deep Learning Phenomena" field that are still not understood. I believe that any steps towards advancing our understanding on this core problem will be very useful in future. And I think the paper does make such steps: (a) many reviewers found the CIFAR/MNIST interesting, (b) the authors explicitly demonstrate that for their specific problems DNNs indeed exhibit strong biases, and moreover these biases are quite surprising (for instance, Observation 2). That being said, there were several concerns raised by the reviewers. I require the authors to account for them and revise the draft accordingly. The following list contains some of them (while the authors should also refer to the detailed reviews to address all other minor ones as well): (1) The writing of the paper in many places is overly bold. The authors use loud words such as "explain" multiple times. In some places they speculate on implications for practical settings. The authors should clarify where the results are context-specific. The authors should avoid speculating about robustness (they seem to claim that the simplicity bias is the major reason behind non-robustness, while this claim is not supported at all). The authors should consider moving Section 4.3 to the "Discussions" section, clearly emphasizing the argument is speculative. (2) The authors should clearly mention that for any given bias one can construct a problem where this bias hurts. This is essentially the argument of Rev#3. Therefore, it is dangerous to conclude that "the bias hurts generalization of DNNs in the natural tasks" from the fact that "the bias hurts DNNs in the specially designed synthetic problems". The authors have no evidence that the implicit bias of DNNs (whatever it is) hurts their generalization when trained, say, on natural image datasets. Moreover, for natural images, the concept of "feature" in "simple feature" is not clear: is it a pixel? A patch? Or the output of the filter? (3) Rev#2 raised several good questions. First, the authors seem to blame mainly SGD for the simplicity bias. But, perhaps, DNN architecture and loss function may also play a role? Second, it would be nice to discuss the influence of the initialization scheme on the observed simplicity bias, specifically the role of the initialization scale (i.e. the magnitude of the weights in the beginning of training). It may be that if we use a larger scale, DNNs will pick up the "complex" features. On top of this, I will ask the authors to clarify the role of $d$ in Theorem 1 and more generally in the paper: (a) The proof of Theorem 1 assumes (at some point, see Lemma F3, for instance) that $d > exp( (8c/eta)^2)$. If we set learning rate to the unrealistically large value eta=1, then we get $d > exp( (8 * 10)^2)$. How come this *extremely strong* assumption was not mentioned in the main part of the paper? (b) Moreover, it seems that everywhere in the paper experiments use a *large number* of features (50). So, I feel, the empirical/theoretical evidence of this paper supports the fact that "when there are *many* complex and *one* simple feature DNN picks up the simple one" with emphasize on *many*. What happens if we only have 2 features, i.e. d=2? Theorem 1 in this case becomes vacuous. What about empirical results? I ask the authors to explicitly address this question in the revision. I feel all the requirements listed above are realistic and I also believe that addressing them will significantly improve the presentation of the paper. I tend to trust the authors in implementing these modifications. Otherwise, I feel this paper deserves to be published.